# Inhibitory neurons in the superior colliculus mediate selection of spatially-directed movements

Jaclyn Essig[1], Joshua B. Hunt [1] & Gidon Felsen [1]✉

Decision making is a cognitive process that mediates behaviors critical for survival. Choosing spatial targets is an experimentally-tractable form of decision making that depends on the midbrain superior colliculus (SC). While physiological and computational studies have uncovered the functional topographic organization of the SC, the role of specific SC cell types in spatial choice is unknown. Here, we leveraged behavior, optogenetics, neural recordings and modeling to directly examine the contribution of GABAergic SC neurons to the selection of opposing spatial targets. Although GABAergic SC neurons comprise a heterogeneous population with local and long-range projections, our results demonstrate that GABAergic SC neurons do not locally suppress premotor output, suggesting that functional long-range inhibition instead plays a dominant role in spatial choice. An attractor model requiring only intrinsic SC circuitry was sufficient to account for our experimental observations. Overall, our study elucidates the role of GABAergic SC neurons in spatial choice.

[1] Department of Physiology and Biophysics, and Neuroscience Program University of Colorado School of Medicine, Aurora, CO, USA. ✉email: gidon.felsen@cuanschutz.edu

Decision making is fundamental for adaptive behavior and examining its neural bases offers insight into circuit mechanisms for cognitive processes[1,2]. An effective strategy to study decision-making circuits is to examine the activity of specific types of neurons in conserved brain areas during ethologically relevant behaviors[3]. Spatial choice—selecting where to attend or to move—is ideal for this purpose: it is an adaptive form of decision making that is critical for survival and, across primates[4–7], cats[8] and rodents[9], depends on computations performed by the intermediate and deep layers of the midbrain superior colliculus (SC). Behavioral targets in contralateral space are topographically represented in the SC (primates[10–12]; cats[13]; rodents[14–17]) as early as other choice-related brain regions[18,19], and manipulating SC activity during choice produces predictable choice biases (primates[20–23]; rodents[24–28]). These and additional data showing that SC activity reflects target value and other decision-related variables[14,29–31] demonstrate that the SC is required for selecting where to move[4], which we refer to here as spatial choice. While the SC has recently served as an exquisite experimental system for probing how neural circuits contribute to SC-dependent functions including visual attention[28], approach[32] and escape[33,34] behaviors, and wakefulness[35], the circuit mechanisms underlying spatial choice remain unclear. For example, despite SC activity topographically representing contralateral space, a subpopulation of SC neurons in rodents paradoxically exhibits higher activity for ipsilateral targets[14,25]. These data are not accounted for by current SC circuit-level frameworks of spatial choice.

Inhibition is a critical component in several descriptive models of SC decision-making circuits[36–38], but experimental evidence in support of a functional role for inhibitory SC neurons in spatial choice is lacking. Approximately 30% of SC neurons are GABAergic[39,40]. Although inhibitory SC neurons have been posited to mediate spatial choice[36,38,41], inhibition from extra-collicular sources[42,43] may instead play a dominate role. In particular, the substantia nigra pars reticulata provides widespread inhibition to the SC that modulates activity related to movement initiation and target selection[43–49]. Dissociating the role of inhibitory SC neurons in spatial choice from the influence of extra-collicular inhibition has remained a challenge, in large part due to the inability of traditional methods to isolate GABAergic SC neurons for recording and manipulation during behavior.

Further complicating inferences about the role of GABAergic SC neurons in decision making is their morphological diversity and correspondingly diverse physiological properties[39]. GABAergic SC neurons have been shown to locally inhibit neighboring neurons in rodent slice experiments[50–52] and optogenetic activation of GABAergic SC neurons in behaving mice attenuated attention to, or disrupted perception of, contralateral space[28]. Moreover, anatomical and physiological evidence also supports functional long-range inhibition (i.e., inhibition that targets locations distal from the GABAergic cell body, either within the same or opposite SC or in other brain regions) in the SC of rodents[39] and other species[41,53–59]. Together, these data suggest that local and long-range GABAergic projections modulate activity in SC circuits, but little is known about the specific roles of any type of GABAergic SC neuron during behavior, including spatial choice. To elucidate the functions of inhibitory SC neurons in spatial choice, we must first examine how their activity relates to spatial choice in vivo, while targets are selected[3,60].

Previous studies of neural circuit function have used optogenetic activation of GABAergic neurons as a method to locally decrease the activity of neighboring neurons[28,61–64]. When this method was applied to the SC, the results were consistent with the idea that activating GABAergic SC neurons locally decreases the activity of neighboring excitatory neurons[28,62]. We therefore hypothesized that, in the context of spatial choice, GABAergic SC neurons locally suppress nearby glutamatergic premotor output neurons representing contralateral targets[65,66] (Fig. 1b). This hypothesis would also be consistent with the possibility that the subpopulation of SC neurons representing ipsilateral targets are those GABAergic neurons providing local suppression, since their activity would be inversely correlated with premotor output neurons. In this case, we would predict that activation of GABAergic SC neurons during spatial choice (left vs right target) would result in an ipsilateral choice bias (Fig. 1c). However, if local and long-range mechanisms play balanced roles in spatial choice then activating this heterogeneous population of GABAergic SC neurons may result in no consistent behavioral effect.

We examined the functional role of inhibitory SC neurons in spatial choice by manipulating, recording, and modeling GABAergic SC activity in behaving mice. We found that activating GABAergic SC neurons during decision making in a two-alternative spatial choice task produced a contralateral choice bias, which is inconsistent with the idea that GABAergic neurons primarily mediate choice through local suppression of nearby premotor output neurons. We further interrogated these results with cell-type-specific recordings during behavior and revealed a direct relationship between the activity of GABAergic neurons

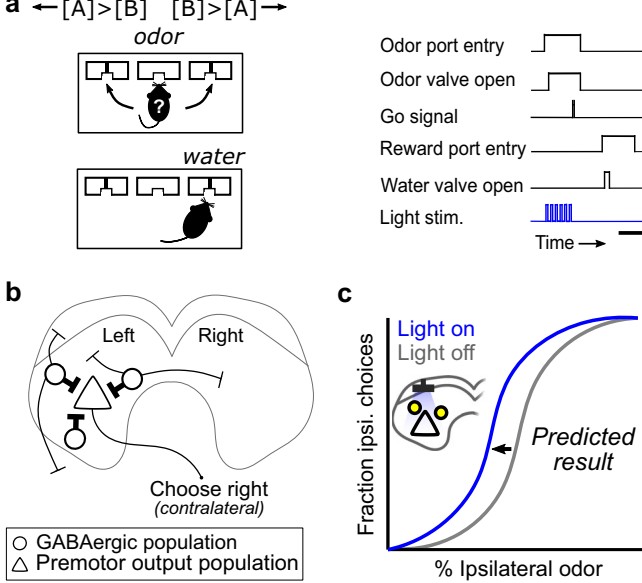

**Fig. 1 Expected functional role for GABAergic SC neurons in a left-vs-right spatial choice task. a** Spatial choice task. Reward location (left or right) was determined by the relative concentrations of Odors A and B [(+)-carvone and (−)-carvone, respectively]. In each trial, the mouse entered the central odor port, chose a reward port while sampling an odor mixture, waited for the go signal, executed its choice, and received water if correct. To test their functional role in spatial choice, GABAergic SC neurons were unilaterally photoactivated during odor sampling on 30% of trials. Scale bar, ~500 ms. **b** GABAergic SC neurons project within one SC, between the SCs and to extracollicular targets. We hypothesized that, during spatial choice, GABAergic SC neurons suppress premotor output activity that promotes rightward (i.e., contralateral) choices. Thicker lines represent higher hypothesized activity during spatial choice. External inputs to the SC are omitted for simplicity. **c** We predicted that photoactivation of GABAergic SC neurons during choice would bias choices ipsilateral to the photoactivated side, indicated by a leftward shift in the psychometric function on light-on trials (blue) compared to light-off trials (gray).

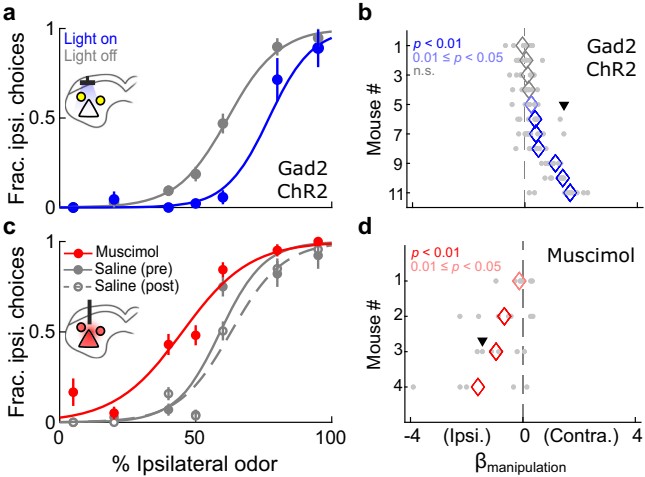

**Fig. 2 Photoactivating GABAergic SC neurons during spatial choice promote contralateral choices. a** Example session in which GABAergic neurons were unilaterally photoactivated during spatial choice. Here, and in all panels, inset schematic shows the experimental setup. Yellow indicates ChR2-YFP expression. **b** Overall influence of GABAergic photoactivation on choice ($p = 0.0029$, Wilcoxon signed-rank test; $n = 7/11$ mice, $p < 0.05$, permutation test), quantified by logistic regression (Methods). See also Supplementary Fig. 2a for comparison to YFP controls and Supplementary Fig. 3a, c for reaction times. Mice are sorted by effect size. Individual sessions are plotted in light gray. Arrowhead, example session in (**a**). **c** Example sessions in which muscimol or saline was unilaterally delivered to the SC prior to the behavioral session. **d** Overall influence of local inhibition on choice in muscimol sessions ($n = 4/4$ mice, $p < 0.05$, permutation test), quantified by logistic regression. Individual sessions are plotted in light gray. Arrowhead, example session in (**c**).

and contralateral choices. Finally, we developed a biologically- and behaviorally-constrained attractor model to examine how, without requiring circuitry extrinsic to the SC, activating a diverse population of GABAergic SC neurons could promote contralateral choices. Overall, we have identified a non-local functional role for inhibitory SC neurons in an important form of decision making.

## Results

**Activating GABAergic SC neurons promotes contralateral choices.** We aimed to identify a functional role for inhibitory neurons in the intermediate and deep output layers of the SC in spatial decision making. GABAergic SC neurons are cytoarchitecturally diverse[39] and likely play several roles in multiple SC-dependent functions[28,32–35], but their role in choice has not been directly examined experimentally. To begin to test the function of inhibitory SC neurons during choice, we unilaterally photoactivated channelrhodopsin-2 (ChR2)-expressing GABAergic SC neurons in Gad2-Cre mice while they selected one of two distinct targets in a spatial choice task (Supplementary Fig. 1a; Methods). Briefly, the task required mice to sample a binary odor mixture that cues reward location (left or right), wait for a go signal, and execute an orienting movement towards the selected location to retrieve a water reward. Spatial choice, the process of selecting a movement to either the left or right reward location (i.e., action selection), occurs as mice sample the odor mixture and wait for the go signal. GABAergic SC neurons were photoactivated on 30% of trials during spatial choice[26,67] (Fig. 1a; Methods). Population activity in the SC represents contralateral targets[5,12,16], including in mice performing this task[17]. We thus

expected that if GABAergic neurons primarily mediate decisions between left and right targets via local inhibition, activating GABAergic SC neurons (a technique commonly used to locally inhibit activity in SC and other circuits[28,62]) during spatial choice would decrease activity of nearby premotor neurons representing the contralateral choice relative to activity in the opposite SC (Fig. 1b), and therefore bias choices ipsilaterally (Fig. 1c). If the local and long-range inhibition provided by GABAergic SC neurons is balanced, though, we would not expect activating them to result in a consistent behavioral effect.

However, photoactivating GABAergic SC neurons during odor sampling resulted in a clear and significant contralateral choice bias (rightward shift in psychometric function on light-on trials (blue) compared to light-off trials (gray); Fig. 2a). This effect was observed across mice ($p = 0.0029$, Wilcoxon signed-rank test) and in a majority of the mice tested ($n = 7/11$ mice, $p < 0.05$, permutation test; Fig. 2b), and differed from the effect exhibited by control mice expressing only YFP ($p = 8.19 \times 10^{-5}$, $U = 3921$, two-tailed Mann–Whitney $U$ test; Supplementary Fig. 2). Furthermore, in sessions with a significant contralateral choice bias ($p < 0.05$, permutation test), we found that reaction times (from go signal to odor port exit) on light-on trials decreased for contralateral choices and increased for ipsilateral choices ($n = 33$ sessions; Ipsi. choices: $p = 0.0049$, $W = 438$; Contra. choices: $p = 3.4 \times 10^{-4}$, $W = 80$, two-tailed Wilcoxon signed-rank test; Supplementary Fig. 3a, c), consistent with a range of models positing a competition between opposing targets in the SC[36,38,68–70].

Since photoactivation of GABAergic SC neurons promoted a contralateral bias, as opposed to the ipsilateral bias expected from local suppression of premotor output neurons (Fig. 1c), we next examined the behavioral effects of local pharmacological inhibition in mice performing the task. Muscimol, a GABA_A receptor agonist, has been used to silence small volumes of SC in rats[24,71] and primates[23,72] performing analogous spatial choice tasks, and results in an ipsilateral choice bias. Consistent with previous reports, and in contrast to the behavioral results from GABAergic activation (Fig. 2a, b), unilateral infusion of muscimol reliably biased choices ipsilaterally (leftward shift in the psychometric function (red) compared to saline sessions (gray) in Fig. 2c) for all mice tested (Fig. 2d; $n = 4/4$ mice, $p < 0.05$, permutation test). These results affirmed that local suppression of SC activity promotes ipsilateral choices in mice performing the task, providing further evidence against the idea that GABAergic SC neurons suppress local premotor output during spatial choice. Instead, the effects of GABAergic activation on choice and reaction time were quite similar to the effects induced by photoactivating glutamatergic SC neurons during spatial choice (Methods; Supplementary Figs. 2c, 3b, c, 4). Overall, the contralateral bias observed by activating GABAergic SC neurons during spatial choice suggests that, despite their anatomical and physiological diversity, they perform a predominantly non-local role in the selection of spatial targets.

**Endogenous GABAergic SC activity corresponds to contralateral choices.** The results of our manipulation experiments prompted us to analyze local neuronal responses to GABAergic photoactivation, and endogenous GABAergic activity in the absence of light, during spatial choice. We employed an optogenetic identification strategy (i.e., optotagging) to target extracellular recordings to GABAergic SC neurons in the same animals used for stimulation/behavior experiments (Supplementary Fig. 5; Methods). GABAergic neurons were identified by the latency of their response to light during stimulation/recording sessions (reliably spiking within 5 ms following each light pulse) and

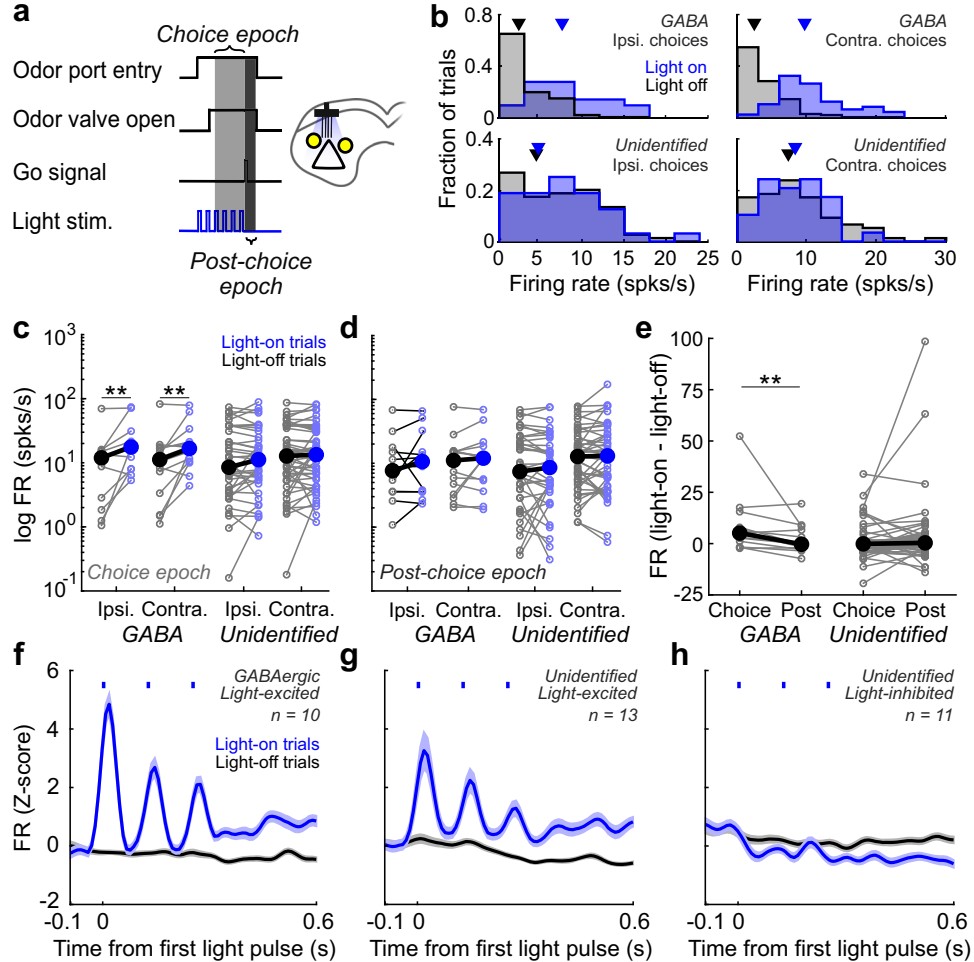

**Fig. 3 Effects of photoactivation are restricted to GABAergic SC neurons during the choice epoch. a** Schematic of experimental paradigm and epoch definitions, in stimulation/behavior/recording sessions. **b** Firing rates on light-on (blue) and light-off (black) trials for example GABAergic and unidentified neurons, separated by choice (ipsi. or contra.). Arrowheads, median firing rate. **c** Effect of photoactivation on local activity in the choice epoch separately for ipsi. and contra. choices. GABAergic: Ipsi. choices, $n = 12$ neurons, **$p = 0.0034$; Contra. choices, $n = 14$ neurons, **$p = 0.004$. Unidentified: Ipsi. choices, $n = 37$ neurons, $p = 0.79$; Contra. choices, $n = 48$ neurons, $p = 0.98$, two-tailed Wilcoxon signed-rank tests. Neurons were analyzed if they met firing rate criteria (Methods). Each neuron is represented by a connected pair of small symbols. Large symbols show population medians. **d** As in (**b**), for activity in the post-choice epoch. GABAergic: Ipsi. choices: $n = 12$ neurons, $p = 0.52$; Contra. choices: $n = 14$, $p = 0.76$. Unidentified: Ipsi. choices: $n = 37$ neurons, $p = 0.52$; Contra. choices: $n = 48$ neurons, $p = 0.36$, two-tailed Wilcoxon signed-rank tests. See also Supplementary Fig. 6. **e** Difference in light response between the choice epoch and post-choice epoch for ipsi. and contra. choices combined for GABAergic neurons (**$p = 0.0017$, Wilcoxon signed-rank test) and unidentified neurons ($p = 0.26$, Wilcoxon signed-rank test). **f** Mean z-scored firing rate for GABAergic neurons that exhibited a significant increase in firing rate on light-on trials during the choice epoch (Light-excited; $n = 10/14$ neurons; Supplementary Table 2). Activity on ipsilateral trials is shown; activity on contralateral trials was similar. Blue ticks, light delivery. Shading, ± SEM. **g** As in (**f**), for unidentified neurons that increase their firing rates (Light-excited; $n = 13/51$ neurons). **h** As in (**f**), for unidentified neurons that decrease their firing rates (Light-inhibited; $n = 11/51$ neurons).

by the correlation between light-driven and spontaneous action potential waveforms ($r^2 > 0.95$; Supplementary Fig. 5e; Methods).

We first sought to confirm that light delivery during spatial choice had the desired effect of directly and transiently increasing the firing rates of GABAergic SC neurons. To do so, we recorded optotagged GABAergic neurons, as well as nearby neurons that did not reliably respond to light and could therefore be glutamatergic or GABAergic ("unidentified neurons"; Supplementary Fig. 5e), in a subset of stimulation/behavior sessions (Fig. 3; $n = 15$ stimulation/behavior/recording sessions; Supplementary Table 1). We then compared single-unit firing rates on light-on trials to light-off trials, separately for ipsilateral and contralateral choices, for the choice epoch (defined as the time from 100 ms after odor valve open until the go signal; Fig. 3a, b; Methods). We found that the activity of our GABAergic

population of neurons increased during light-on trials beyond endogenous activity levels (measured during light-off trials) when mice chose either the ipsilateral target (Fig. 3c; $p = 0.0034$) or the contralateral target ($p = 0.004$, two-tailed Wilcoxon signed-rank tests). Neuron-by-neuron analyses also revealed that 71% of GABAergic neurons exhibited a significant overall increase in firing rate during the choice epoch (Supplementary Table 2; Fig. 3f).

Conversely, light had no overall effect on the activity of our unidentified population of neurons (Fig. 3c; Ipsi. choices: $p = 0.79$; Contra. choices: $p = 0.98$; two-tailed Wilcoxon signed-rank tests), with only 47% of unidentified neurons exhibiting any light-induced change in firing rate, with approximately as many exhibiting an increase (Fig. 3g; $n = 13$; $p < 0.05$) as exhibiting a decrease (Fig. 3h; $n = 11$; $p < 0.05$, Mann–Whitney $U$ tests) (Supplementary Table 2). In addition to confirming that light had

the desired effect of increasing GABAergic activity during the choice epoch, these data also argue against a prominent role for disinhibition during spatial choice. If disinhibition were occurring, we would expect to observe a light-elicited decrease in activity for a subpopulation of GABAergic neurons and a clear increase in activity of unidentified neurons, which was inconsistent with our data (Fig. 3c; Supplementary Table 2).

We next examined whether the contralateral choice bias observed following GABAergic activation (Fig. 2a, b) was the result of an abrupt increase in activity that can occur with prolonged light-induced inhibition (i.e., rebound excitation)[73]. We first tested for signatures of rebound excitation by analyzing activity from our stimulation/recording sessions used for optotagging. We found that a minority of all neurons exhibited significantly decreased activity during photoactivation of GABAergic neurons ($n = 32/301$ neurons; Supplementary Fig. 6a) and there was no increase in the activity of unidentified neurons when comparing firing rates before and immediately after the photoactivation period (Supplementary Fig. 6b). Although these results argue against rebound excitation mediating the contralateral bias we observe when activating GABAergic SC neurons during spatial choice (Fig. 2a, b), these analyses do not account for circuit dynamics in the context of behavior, which may allow for rebound excitation. To ensure rebound excitation was not occurring during the task, activity immediately following the light delivery period (the time from the go signal until odor port exit; "post-choice epoch"; Fig. 3a) was analyzed. We reasoned that rebound excitation would be reflected in higher firing rates immediately following light stimulation on light-on trials than during an equivalent epoch on light-off trials. We found no difference between firing rates of the unidentified population on light-on and light-off trials (Fig. 3d; Ipsi. choices: $p = 0.52$; Contra. choices: $p = 0.36$; two-tailed Wilcoxon signed-rank tests), nor was there a difference in the GABAergic population (Fig. 3d; Ipsi. choices: $p = 0.52$; Contra. choices: $p = 0.76$, two-tailed Wilcoxon signed-rank tests), suggesting that rebound excitation cannot account for our behavioral results. Finally, we compared the overall effect light had on the firing rate between the choice and post-choice epochs and found that light had a greater effect during the choice epoch than the post-choice epoch for GABAergic neurons (Fig. 3e; $p = 0.0017$) but not for unidentified neurons (Fig. 3e; $p = 0.26$, Wilcoxon signed-rank tests). These results demonstrate that light delivery exclusively increased firing rates in GABAergic neurons during the choice epoch and confirm the relationship between increased GABAergic SC activity and increases in contralateral choice (Fig. 2a, b).

However, these photoactivation experiments do not offer insight into normally occurring GABAergic SC activity during spatial choice. We therefore examined the endogenous activity of these neurons during spatial choice, in the absence of photoactivation. We sought to determine if activity correlates with contralateral choices, which would be consistent with the manipulation results (Fig. 2a, b) or correlates with ipsilateral choices, which would be consistent with the hypothesis that GABAergic SC neurons locally suppress premotor output activity during spatial choice (Fig. 1b). By performing our optotagging analysis in conjunction with each behavior/recording session (Supplementary Table 1), we conservatively identified 74 GABAergic neurons from a total recorded population of 239 SC neurons that exhibited sufficient activity during the choice epoch (Supplementary Fig. 5; Methods) and examined choice-epoch activity separately during ipsilateral and contralateral choices. Figure 4a highlights an example GABAergic neuron showing a greater increase in firing rate during contralateral than ipsilateral choices. We quantified the dependence of firing rate on

choice for all recorded neurons by calculating the choice preference of each neuron during the choice epoch[74–76] (Methods). This ROC-based analysis uses a sliding criterion to define a curve separating firing rates associated with ipsilateral and contralateral choices (Fig. 4b). The resulting values range from −1 (greater activity before ipsilateral choices) to 1 (greater activity before contralateral choices), with 0 indicating no detectable difference in activity. Across all SC neurons with a significant choice preference, more exhibited a contralateral than ipsilateral preference (Fig. 4c, Top; Contra.-preferring neurons: 49/72; Ipsi.-preferring neurons = 23/72; $p = 0.0022$, $X^2$ test), consistent with the previous results[14,17,25,77]. We also found that more neurons preferred the contralateral than the ipsilateral choice within the population of identified GABAergic neurons (Fig. 4c, Bottom; Contra.-preferring neurons = 25/29, Ipsi.-preferring neurons = 4/29; $p = 9.7 \times 10^{-5}$, $X^2$ test). Psychometric functions constructed conditionally on the endogenous choice-epoch firing rate of contralateral-preferring GABAergic neurons further demonstrate that contralateral choices were more likely on trials with higher firing rates (Fig. 4d). Finally, during the choice epoch, we observed diversity in the timing of peak activity across the population of contralateral-preferring GABAergic neurons (Fig. 4e), with some exhibiting maximal activity early in the epoch and others just before the movement was permitted. These results demonstrate that the preponderance of GABAergic SC neurons that exhibit a preference reflect contralateral choice, consistent with the contralateral bias elicited by activating the entire population of GABAergic SC neurons during spatial choice.

We applied similar choice-epoch analyses to the endogenous activity of unidentified light-inhibited neurons recorded in stimulation/behavior/recording sessions (Fig. 3h) to further examine if they were likely to be premotor output neurons. The choice-epoch activity of these neurons on light-off trials was higher for ipsilateral than contralateral choices (Supplementary Fig. 7b), opposite of what we would expect from premotor output neurons (which typically exhibit increased activity before contralateral choices) and in contrast to the choice-epoch activity exhibited by other unidentified neurons (e.g., light-excited; Supplementary Fig. 7a, c). These results further demonstrate that, although GABAergic SC neurons provide local inhibition to a subset of nearby neurons during spatial choice, these neurons are unlikely to be premotor output neurons.

We next utilized the stimulation/behavior/recording sessions to examine how tightly GABAergic SC activity and spatial choices are coupled. Specifically, we leveraged the fact that photoactivation elicits a variable number of spikes to test whether the likelihood of contralateral choice on a given trial depended on the number of spikes added during the choice epoch (Fig. 5a). We reasoned that contralateral choices would be more likely to occur on trials with more light-elicited spikes—particularly on difficult trials in which the sensory evidence indicating the reward side was weak[9,23,28,72]. Figure 5b shows an example GABAergic neuron in which contralateral choices were more likely on trials with more light-induced spikes, as demonstrated by a positive exponential fit to trial-by-trial data (Methods). We performed this analysis for each contralateral-preferring GABAergic neuron recorded in stimulation/behavior/recording sessions ($n = 5$), separately for difficult (% Ipsi. odor = 40, 50, or 60) and easy (% Ipsi. odor = 5, 20, 80, or 95) trials. Indeed, there was a stronger relationship between added spikes and choice direction on difficult than on easy trials (Fig. 5c; $p = 0.0268$, two-tailed paired $t$-test), further supporting the functional role of GABAergic SC activity in promoting contralateral choices.

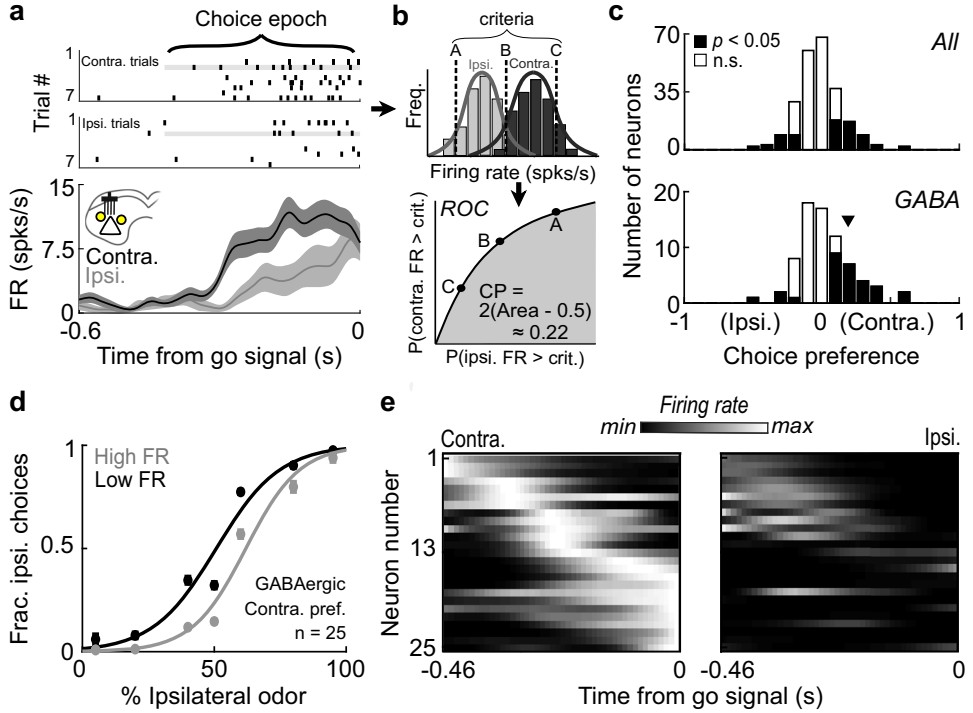

**Fig. 4 Endogenous activity of GABAergic SC neurons corresponds to contralateral choices. a** Rasters and peri-event histograms for an identified GABAergic neuron recorded in a behavior/recording session aligned to go signal and segregated by choice direction. Rasters show seven randomly selected trials per group. Shading, ±SEM. **b** Calculation of choice preference for neurons recorded in behavior/recording sessions. Distributions of trial-by-trial choice-epoch firing rates are calculated across ipsilateral and contralateral choices, from which an ROC curve is constructed by computing the fraction of trials, separately for ipsilateral and contralateral choices, with higher firing rates than a shifted criterion (e.g., A, B, C). Choice preference ranges from −1 (strongest ipsilateral preference) to 1 (strongest contralateral preference). **c** Choice preferences for all neurons and for GABAergic neurons only. Black, neurons with significant choice preference ($p < 0.05$). All neurons: 49 prefer contra., 23 prefer ipsi., $p = 0.0022$, $X^2$ test; GABAergic neurons: 25 prefer contra., 4 prefer ipsi., $p = 9.7 \times 10^{-5}$, $X^2$ test. Arrowhead, example neuron in (**a**). **d** Conditional psychometric functions constructed separately for trials in which endogenous choice-epoch firing rate was high (top quartile) or low (bottom quartile) for GABAergic neurons with a significant contralateral choice preference (right black bars at bottom of (**c**)). **e** Normalized activity during the choice epoch of each contralateral-preferring GABAergic neuron (right black bars at bottom of (**c**)), separated by choice on each trial. Neurons are sorted by time of maximum firing rate for contralateral choices.

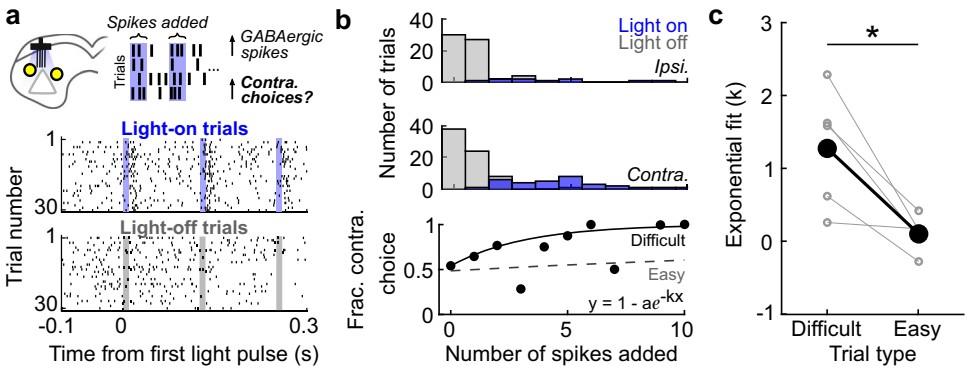

**Fig. 5 Contralateral choice is more likely on trials in which light elicits more spikes in GABAergic SC neurons. a** Schematics of experimental set-up and analysis, and rasters for an example contralateral-preferring GABAergic neuron recorded in a stimulation/behavior/recording session, aligned to first light pulse separately on contralateral light-on and light-off trials. Shading indicates the timing of light pulses on light-on trials. **b** Histograms show number of ipsilateral and contralateral choices as a function of number of spikes during time of light delivery on light-on trials and matched time on light-off trials. Circles show fraction of contralateral choices for each number of spikes added. Solid line, best-fit exponential function to raw trial-by-trial values (ipsilateral choice = 0; contralateral choice = 1) for difficult trials (% Ipsi. odor = 40, 50, or 60). Dashed line, same fit for easy trials (% Ipsi. odor = 5, 20, 80, or 95). **c** For each GABAergic neuron with a significant contralateral choice preference recorded in stimulation/behavior/recording sessions ($n = 5$), slope of exponential fit ($k$) is plotted for difficult and easy trials (*$p = 0.027$, two-tailed paired $t$-test). For this analysis neurons were included in our GABAergic population if they responded within 7 ms of photostimulation (Methods). Large symbols show population means.

**Attractor model is consistent with intercollicular inhibition outweighing local inhibition in mediating spatial choice.** Together, the results of the manipulation and recording experiments argue against the hypothesis that GABAergic SC neurons contribute to spatial decision making by locally suppressing premotor output, suggesting instead a role for non-local functional inhibition. GABAergic SC neurons project both to distal SC regions (in the same and opposite SC) and to extracollicular structures[39]. To determine if mechanisms within the two SCs (i.e., without invoking extracollicular structures) are sufficient to reproduce a contralateral choice bias when local and long-range inhibitory neurons are photoactivated, we utilized a rate-based bump attractor model[17,78] (Methods) similar to those used to elucidate other functions of the SC such as working memory[27] and countermanding behavior[19].

In our model the populations of excitatory (E) and inhibitory (I) cells (in a 2:1 ratio) in each colliculus are interconnected using specific intracollicular (intra-SC) distance-dependent rules, where I cells can influence neural activity at overall greater distances than E cells (Fig. 6a). A small subpopulation of E and I cells also form sparse intercollicular (inter-SC) connections, consistent with experimental observations in mice and other species[39,57,58,79].

We initially selected model parameters for which a sustained locus of activity emerged following external stimulation of E and I cells, which was stabilized by local interactions. On each model trial, the choice was read out as the side contralateral to the SC with the highest mean E cell activity at the end of the choice epoch. This model allowed us to test, over a range of valid parameters, the effect of manipulating the activity of specific cell types, and thereby determine which GABAergic architectures were consistent with the experimental data.

To model photoactivation of GABAergic neurons (Fig. 2a, b), we unilaterally increased the rate of a subset of I cells for the equivalent of ~500 ms (the length of the choice epoch in the task) on 30% of trials in each session (Methods). We fit the model such that manipulation reproduced the contralateral choice bias observed in our experiments (Fig. 6b, c). Using the same parameters, we next modeled pharmacological SC inhibition (Fig. 2c, d) by unilaterally decreasing the activity of a subset of E and I cells, and photoactivation of glutamatergic SC neurons (Supplementary Fig. 4) by unilaterally increasing the rate of a subset of E cells. For both sets of experiments, the model frequently reproduced the choice bias observed experimentally (Fig. 6d–g).

We next examined how these results depended on specific model parameters. We found that the contralateral choice bias induced by exciting I cells (Fig. 6b, c) depended on the synaptic strength of the intercollicular inhibitory (inter-I) connections: low values of this parameter yielded an ipsilateral choice bias, while high values yielded a contralateral choice bias (Fig. 6h, black symbols). In contrast, inter-I strength did not systematically modulate the effects of exciting E cells or inhibiting all cells (Fig. 6h, gray symbols). Importantly, the absolute inter-I strength did not entirely determine the direction of the choice bias under I cell excitation; rather the ratio of inter-I to intracollicular inhibition (intra-I) strength was critical: exciting I cells resulted in the experimentally observed contralateral choice bias when the inter-I strength was at least 2.5 times higher than the intra-I strength (Fig. 6i). Overall, the ratio of inter-I to intra-I strength provides one explanation for how photoactivating a heterogeneous population of GABAergic neurons with local and long-range projections could promote contralateral choices (Figs. 2a, b, 5).

## Discussion
Elucidating the roles of specific cell types in decision making is a critical step toward understanding functional neural circuits for decisions. In the SC, while the functions of some types of neurons have been studied during behavior[32–35], characterization of inhibitory neurons in the intermediate and deep layers has primarily been limited to anatomy and slice physiology[39,40,51,52,79]. In this study, we leveraged behavioral, optogenetic, electrophysiological, and computational approaches to examine a functional role of GABAergic SC neurons in spatial choice, an important and tractable form of decision making for which the SC plays a critical role conserved across species[4,9]. Unilateral manipulation of, and extracellular recordings from, GABAergic SC neurons during spatial choice revealed that GABAergic neurons promote contralateral choices (Figs. 2b, c, 4, 5). We further explored intrinsic SC mechanisms that could account for our behavioral results using an attractor model requiring only intrinsic SC circuitry (Fig. 6). The model suggested that contralateral choices could be promoted by activating local and long-range GABAergic neurons if the functional strength of intercollicular GABAergic SC neurons outweighs the contribution of locally projecting GABAergic SC neurons (Fig. 6i). Overall, these results offer insight into cell-type-specific mechanisms mediating spatial choice and extend our understanding of inhibition in decision-making circuits[19,27,36,38,80–82].

In particular, our findings provide evidence against the idea that GABAergic SC neurons locally suppress premotor output during spatial choice (Fig. 1b). While we found that a population of SC neurons was locally inhibited when GABAergic SC neurons were activated during spatial choice (Fig. 3h, Supplementary Table 2), these neurons exhibited higher activity for ipsilateral than contralateral choices (Supplementary Fig. 7b), inconsistent with the representation of contralateral targets by premotor output neurons. These locally inhibited neurons, and not GABAergic SC neurons, may comprise the subpopulation of ipsilaterally preferring neurons that we and others have observed[14,25] (Fig. 4c). GABAergic SC neurons instead may integrate the activity of contralateral-preferring SC neurons[83], along with external input from the basal ganglia and other structures[45,84], to influence choice via functional long-range inhibition of distal populations (Fig. 7). Future experiments can build on ours by identifying the functional roles in spatial choice of GABAergic SC-recipient regions; in particular, recording and manipulating the activity of intercollicular neurons during spatial choice can test the predictions of our attractor model.

The model suggests that one potential target of long-range inhibition consistent with our results is the opposite SC. While GABAergic SC neurons are known to directly project between the two SCs[39,79], other mechanisms may also mediate intercollicular interactions, including multi-synaptic, or extracollicular, pathways, as has been described in the barn owl optic tectum (homolog to the mammalian SC)[80]. Our model also does not rule out a role in spatial choice for long-range projections of GABAergic SC neurons to other brain regions[39], since it was explicitly designed to examine how our results could be explained with only SC circuitry. Notably, the SC is bidirectionally connected with several cortical regions known to be involved in decision making. In rodents, the roles in spatial choice of the frontal orienting field (FOF), the anterior lateral motor cortex (ALM), and anterior cingulate cortex (ACC) have been studied in similar contexts as in the current study[27,61,85]. Interestingly, similar to the SC, unilateral inhibition of ALM principal neurons during action selection disrupts contralateral choices[61]. However, if inhibition of ALM is limited to early in the selection process, circuit dynamics can be recovered by activity from the opposite ALM and accurate action selection is preserved[86], while similar early inhibition of SC disrupts choice[27]. Thus, the two ALM hemispheres appear to cooperate, while the two sides of the SC appear to compete, to mediate left-vs-right spatial choice.

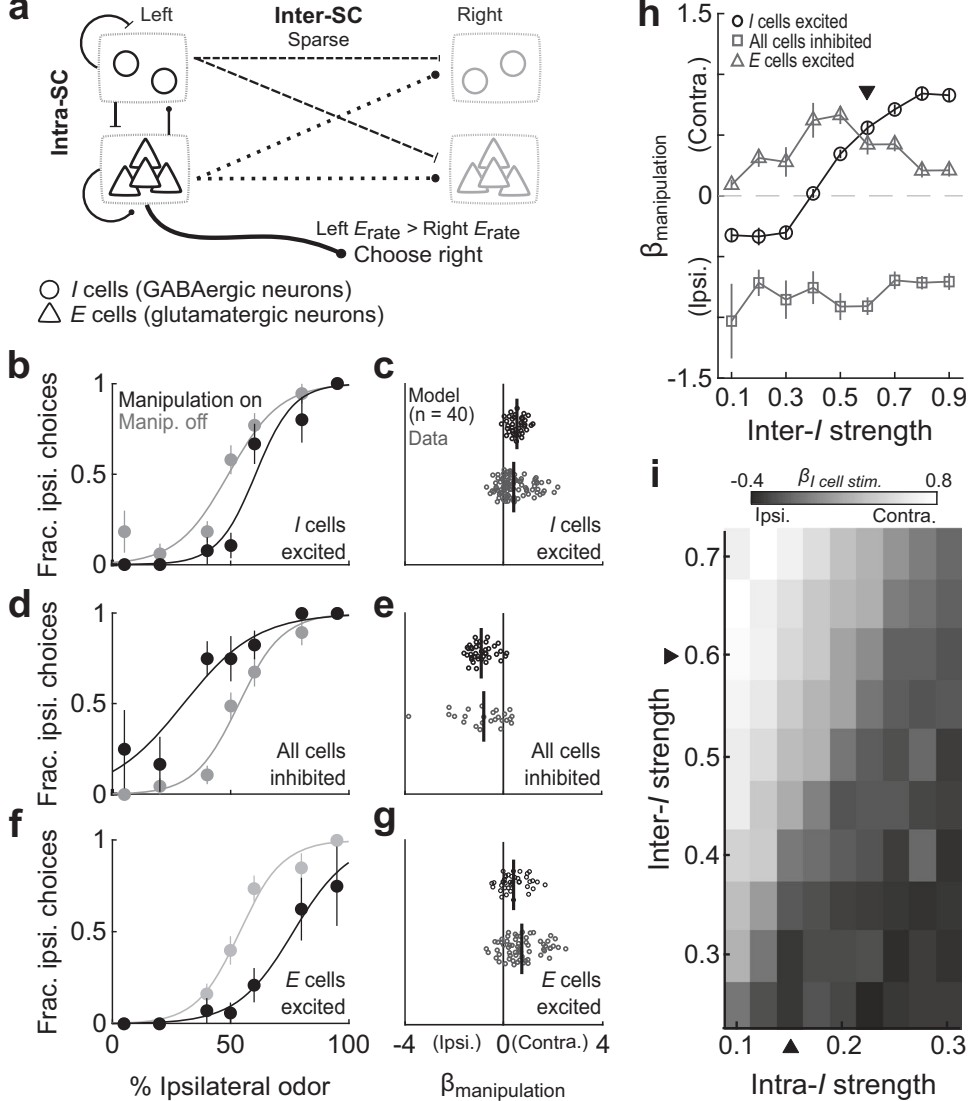

**Fig. 6 The relative weight of local and intercollicular inhibition can explain the contralateral choice bias produced by photoactivation of GABAergic SC neurons. a** Attractor model schematic. **b** Unilateral excitation of inhibitory (*I*) cells during spatial choice in the model (compare to Fig. 2a). **c** Effect of unilaterally exciting *I* cells in model sessions ($n = 40$, $p < 0.0001$, Wilcoxon signed-rank test) and of experimental photoactivation of GABAergic neurons (data in Fig. 2b). Black line, mean effect across sessions. **d** Unilateral inhibition of *E* and *I* cells in the model (compare to Fig. 2c). **e** Effect of unilaterally inhibiting *E* and *I* cells in model sessions ($n = 40$, $p < 0.0001$, Wilcoxon signed-rank test) and of experimental local inhibition (data in Fig. 2d). Black line, mean effect across sessions. **f** Unilateral excitation of *E* cells during spatial choice in the model (compare to Supplementary Fig. 4a). **g** Effect of unilaterally exciting *E* cells in model sessions ($n = 40$, $p < 0.0001$, Wilcoxon signed-rank test) and of experimental photoactivation of glutamatergic neurons (data in Supplementary Fig. 4b). Black line, mean effect across sessions. **h** Effect of manipulations in model sessions depends on strength of intercollicular projections of *I* cells (inter-*I*). Each symbol shows mean ± SEM. Arrowhead, strength in (**b**–**g**). **i** Effect of exciting *I* cells depends on relative strength of their intercollicular (inter-*I*) and intracollicular (intra-*I*) projections. Each element shows mean across 40 sessions. Arrowheads, strengths in (**b**–**g**).

Understanding how SC circuits mediate spatial choice will allow us to examine how the network of regions with which it is interconnected coordinate to perform more complex cognitive functions.

A recent study found that unilateral photoactivation of GABAergic SC neurons affected performance on a visual detection task for stimuli presented to the contralateral, but not ipsilateral, visual hemifield[28]. These results are consistent with local suppression of neuronal activity representing contralateral space—the authors' intended, and our hypothesized, effect of photoactivating GABAergic SC neurons—raising the question of why we obtained a seemingly opposite effect on spatial choice (a contralateral choice bias) using essentially the same photoactivation method. Although our study was superficially similar to that of Wang

et al.[28]—e.g., both examined the role of the SC in mice performing a decision-making task—differences between how the tasks engage SC circuits are likely critical. First, the task used by Wang et al.[28] was a visual change detection task, and photoactivation had the largest effect when delivered during the time when visually evoked activity in the SC was highest. It is possible that the same population of SC neurons that we observed to be locally inhibited by GABAergic SC activation (Fig. 3h, Supplementary Table 2), but which were not responsible for contralateral choice (Supplementary Fig. 7b), are required for visual change detection. GABAergic SC neurons project to the superficial SC and may suppress visual responses[39,83,87], providing another potential pathway for disrupting contralateral visual representations. Second, the task used in Wang et al.[28] did not require the selection of a spatial target for

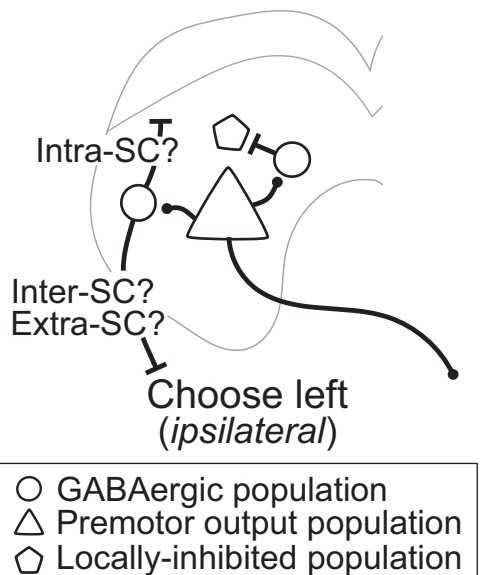

**Fig. 7 Functional long-range inhibition from GABAergic SC neurons promotes contralateral choices.** Long-range projections by which GABAergic SC neurons may promote contralateral choices.

a movement; mice reported a stimulus change in either visual hemifield with a single motor output. If the authors had instead required the selection of a directional movement based on a visual stimulus, or if our study had required a non-directional response to a lateralized stimulus, it is possible that the results of the two studies would have been similar. Future studies are required to address this possibility.

Despite exhibiting no difference in firing rate between easy and difficult trials (Supplementary Fig. 8), activating GABAergic neurons had a stronger influence on choice on difficult trials compared to easy trials (Fig. 5c). Psychometric functions conditioned on endogenous GABAergic SC activity during the choice epoch (Fig. 4d) show a similar relationship: the difference in choice between low and high activity trials is greater for more difficult trials (% Ipsi. odor = 40, 50, or 60). This effect could reflect the balance between intrinsic SC processing and influences extrinsic to the SC such as the basal ganglia and cortical regions, among others, that are known to contribute to spatial choice[9]. When the sensory evidence for the rewarded choice is strong (i.e., on easy trials), input from upstream regions may dominate processing in the SC (or outweigh SC contributions at downstream circuits in the brainstem). Conversely, when the sensory evidence is weak (i.e., on difficult trials), intrinsic SC processing, including via the GABAergic SC neurons studied here, may have more weight in determining the output of competing choices and therefore exert a stronger influence over spatial choice[17].

We have focused on functional SC circuitry mediating a goal-directed spatial choice between two distant targets, each represented in the opposite SC. Consistent with a greater role for the lateral SC in approach behavior[88,89], photoactivation produced a stronger contralateral choice bias when the light was delivered to GABAergic neurons in the lateral SC (Supplementary Fig. 9a). In behavioral contexts outside of spatial choice, GABAergic SC neurons undoubtedly function to shape SC output. In fact, most of the GABAergic neurons we recorded from were indifferent to choice (45/74; Fig. 4c); while these neurons may nevertheless contribute to choice, they may also be involved in acquiring targets or other SC-mediated behaviors not examined in our paradigm. Future studies can expand on the generality of long-range inhibitory mechanisms in decision making by examining

the role of GABAergic SC neurons in other forms of SC-dependent decisions and behaviors, including approach vs. avoidance behavior[88,90], setting criteria for stimulus detection[29], and perceptual sensitivity[28]. While our study examined a heterogeneous population of GABAergic SC neurons, it is likely that different types of GABAergic SC neurons perform specific functions. A finer dissection of inhibitory circuits for decision making and other functions will be feasible as genetic markers for subtypes of GABAergic SC neurons are identified[52,54,91] and leveraged to record and manipulate their activity during behavior.

## Methods

**Animals**. All procedures were approved by University of Colorado School of Medicine Institutional Animal Care and Use Committee. Mice were bred in the animal facilities of the University of Colorado Anschutz Medical Campus or purchased (Jackson Labs). For all experimental conditions, adult male mice (6–12 months old at the time of experiments) from a C57BL/6J background were used in this study. Mice were housed individually in an environmentally controlled room, kept on a 12-h light/dark cycle, and had ad libitum access to food. Mice were water restricted to 1 ml/day and maintained at 80% of their adult weight. Heterozygous Gad2-ires-Cre (Gad2-Cre; Gad2$^{tm2(cre)Zjh/J}$) mice ($n = 11$) were used for optogenetic activation and identification of GABAergic neurons (Supplementary Table 1). Homozygous Vglut2-ires-Cre (Vglut2-Cre; Slc17a6$^{tm2(cre)Lowl/J}$) mice ($n = 4$) were used for optogenetic activation of glutamatergic neurons. Muscimol experiments were performed in C57BL/6J wild-type mice ($n = 4$).

**Behavioral task**. Mice were trained on a previously published odor-guided spatial-choice task[26,67]. Briefly, water-restricted mice self-initiated each trial by nose poking into a central port. After a short delay (~200 ms), a binary odor mixture was delivered. Mice were required to wait $500 \pm 55$ ms (mean ± SD) for a go signal (a high-frequency tone) before exiting the odor port and orienting toward the left or right reward port for water (Fig. 1a). We refer to the time between odor valve open and the go signal as the choice epoch (Figs. 3a, 4a). Exiting the odor port prior to the go signal resulted in the unavailability of reward on that trial, although we still analyzed these trials if a reward port was selected. All training and experimental behavioral sessions were conducted during the light cycle.

Odors were comprised of binary mixtures of (+)-carvone (Odor A) and (−)-carvone (Odor B) (Acros), commonly perceived as caraway and spearmint, respectively. In all sessions—including training on the task, as well as during neural recording and manipulation—mixtures in which Odor A > Odor B indicated reward availability at the left port, and Odor B > Odor A indicated reward availability at the right port (Fig. 1a). When Odor A = Odor B, the probability of reward at the left and right ports, independently, was 0.5. The full set of Odor A/Odor B mixtures used was 95/5, 80/20, 60/40, 50/50, 40/60, 20/80, 5/95. Mice completed training in 8–12 weeks and were then implanted with a neural recording drive, optic fiber, or drug-delivery cannula as described below. All neural recording and manipulation experiments were performed in mice that were well-trained on the task.

**Stereotactic surgeries**. Mice were removed from water restriction and had ad libitum access to water for at least one week before surgery. Preparation for surgery was similar for viral injections and chronic implants. Deep anesthesia was induced with 2% isoflurane (Priamal Healthcare Limited) in a ventilated chamber before being transferred to a stereotaxic frame fitted with a heating pad to maintain body temperature. A nose cone attachment continuously delivered 1.3–1.6% isoflurane to maintain anesthesia throughout the surgery. Scalp fur was removed using an electric razor and ophthalmic ointment was applied to the eyes. The scalp was cleaned with betadine (Purdue Products) and 70% ethanol before injecting a bolus of topical anesthetic (150 μl 2% lidocaine; Aspen Veterinary Resources) under the scalp. The skull was exposed with a single incision and scalp retraction. The surface of the skull was cleaned with saline and the head was adjusted to ensure lambda was level with bregma (within 150 μm). Immediately following all surgeries, mice were intraperitoneally administered sterile 0.9% saline for rehydration and an analgesic (5 mg/kg Ketofen; Zoetis). A topical antibiotic was applied to the site of the incision and mice were given oxygen while waking from anesthesia. Post-operative care, including analgesic and antibiotic administration, continued for up to 5 days after surgery and mice were closely monitored for signs of distress. In addition, mice recovered after surgery with ad libitum access to water for at least 1 week before being water-restricted for experiments.

For viral injections, a small craniotomy was drilled above the left SC (3.64–4.04 mm posterior to bregma, 0.75–1.25 mm lateral of midline, 0.85–2.17 mm dorsal from the brain surface[92]). To maximize overlap between ChR2 expression and the optetrode in Gad2-Cre mice, up to three injections were made within 0.2 mm$^2$. Viruses were delivered with a thin glass pipette at an approximate rate of 100–200 nl/min via manual pressure applied to a 30 ml syringe. Pipets remained at depth for 10 min following each injection before retraction from the brain. For optogenetic manipulation and identification of GABAergic neurons,

Gad2-Cre ($n = 11$) mice were injected with a total (across all injections) of 200 nl of DIO-ChR2-eYFP (AAV2.Ef1α.DIO.ChR2.eYFP, UNC Vector Core, $4.2 \times 10^{12}$ ppm). Vglut2-ires-Cre ($n = 4$) mice were injected with 200 nl of the same virus to express ChR2 in glutamatergic neurons. Control mice expressing only YFP (Gad2-Cre ($n = 2$) and Vglut2-Cre ($n = 2$)) were injected with 100 nl DIO-eYFP (AAV2.Ef1α.DIO.eYFP, UNC Vector Core, $4.6 \times 10^{12}$ ppm). After injection, the skin was sutured (except in the Vglut2-Cre and control mice; see below) and mice recovered for 1 week before being water restricted for behavioral training. Expression occurred during the ~10 weeks of training.

In Vglut2-Cre and control mice, immediately following virus injection, a 105 μm diameter optic fiber (Thorlabs) was implanted in the same craniotomy used for virus injection. The fiber was housed in a ceramic ferrule (Precision Fiber Products MM-FER2007C) and slowly lowered to be slightly (~200 μm) above the injection site. The fiber was affixed to the skull via a single skull screw, luting (3 M), and dental acrylic (A-M Systems).

To deliver light to, and extracellularly record from, the same population of SC neurons, an optetrode drive, an optic fiber surrounded by four tetrodes[93], was chronically implanted above the DIO-ChR2-eYFP injection site in fully trained Gad2-Cre mice. A large (~1 mm²) craniotomy was made around the initial injection site. Three additional small craniotomies were made anterior to the initial injection site: one for implanting a ground wire and two for skull screws. The drive was slowly lowered into the large craniotomy and secured in place with luting and dental acrylic.

To infuse muscimol into the intermediate and deep layers of the SC, a steel guide cannula and removable steel insert assembly (Invivo1) was targeted to 0.8 mm dorsal to the surface of the SC in wild-type mice ($n = 4$). The guide cannula was affixed to the skull with one skull screw, luting, and dental acrylic.

**Optogenetic photoactivation**. A diode-pumped, solid-state laser (473 nm; Shanghai Laser & Optics Century) delivered light through a 105-μm/0.22 numerical aperture patch cable (Thorlabs) attached, prior to each session, to the implanted ferrule or optetrode drive using a ceramic sleeve (Precision Fiber Products SM-CS125S) and index matching solution (Thorlabs G608N3). Power was calibrated with an optic meter before each session (ThorLabs). To activate ChR2+ GABAergic neurons in Gad2-Cre mice during behavior ("stimulation/behavior"; Fig. 2a, b; Supplementary Table 1; $n = 96$ sessions; 11 mice), light was delivered in 10 ms pulses at either 8 Hz (68 sessions) or 25 Hz (28 sessions) during the entire duration of the odor sampling epoch (~500 ms; odor port entry to go signal). We used a range of irradiances extremely unlikely to produce off-target effects (e.g., tissue heating)[94] and did not induce overt movements upon photoactivation (see Supplementary Table 3): 4.4–88 mW/mm² (68 sessions), 109–223 mW/mm² (15 sessions), and 309–427.2 mW/mm² (13 sessions). The behavioral effects observed did not depend on power (Supplementary Fig. 10a) or stimulation frequency used ($p = 0.8815$, $U = 3279$, two-tailed Mann–Whitney $U$ test). Different populations of GABAergic neurons were intentionally stimulated by advancing the optetrode through the SC between sessions, which may have contributed to variability in mice across sessions (Supplementary Fig. 9b). To further confirm that our results were not due to off-target effects, Gad2-Cre mice expressing only eYFP (Supplementary Fig. 2a; $n = 63$ sessions; 2 mice) were stimulated at the highest range of laser powers used for the experimental animals: 257.6 mW/mm² (13 sessions) and 441.6 mW/mm² (50 sessions) at 8 Hz (10 ms on) during the odor sampling epoch.

To activate ChR2+ glutamatergic neurons during the odor sampling epoch in Vglut2-Cre mice (Supplementary Fig. 4; $n = 64$ sessions; 4 mice), the light was delivered at 8 Hz (10 ms on) at irradiances between 3.7 mW/mm² and 29.4 mW/mm². Higher powers (over 50 mW/mm²) produced overt orienting movements of the head in Vglut2-ChR2 mice, and our behavioral effect depended on power (Supplementary Fig. 10b), in contrast to our findings with Gad2-ChR2 mice. Again, to confirm that light delivery did not elicit off-target effects, Vglut2-Cre mice expressing only eYFP (Supplementary Fig. 2c; $n = 52$ sessions; 2 mice) were stimulated using laser powers that exceeded those used for experiments: 55.2 mW/mm² (30 sessions) and 441.6 mW/mm² (22 sessions) at 8 Hz (10 ms on) during the odor sampling epoch. In all Vglut2-Cre and all Gad2-Cre control mice, the same site was stimulated for all sessions. Across all mice and sessions, stimulation occurred randomly on 30% of trials. Extracellular recordings were acquired during 15 photoactivation sessions in Gad2-Cre mice ("stimulation/behavior/recording"; Figs. 3, 5; Supplementary Fig. 7; Supplementary Table 1).

**Muscimol infusion**. Prior to each session, an injection cannula was prepared with either muscimol or saline and inserted into the chronically implanted guide cannula while mice were lightly anesthetized. An infusion pump (Harvard Apparatus) was used to administer 300 nl of solution at 0.15 μl/min. Muscimol dosage was 0.1 mg/ml (in saline) and did not induce ipsilateral circling behaviors. After infusion, the internal cannula remained in place for 3 min before it was retracted. Mice recovered from anesthesia in their cage for at least 10 min before beginning the behavioral session.

**Electrophysiology**. Extracellular neuronal recordings were collected using four tetrodes (a single tetrode consisted of four polyimide-coated nichrome wires

(Sandvik; single-wire diameter 12.5 μm) gold plated to 0.25–0.3 MΩ impedance). Electrical signals were amplified and recorded using the Digital Lynx S multi-channel acquisition system (Neuralynx) in conjunction with Cheetah data acquisition software (Neuralynx).

To sample independent populations of neurons, the tetrodes were advanced between 6–23 h before each recording session. To estimate tetrode depths during each session we calculated distance traveled with respect to rotation fraction of the thumb screw of the optetrode drive. One full rotation moved the tetrodes ~450 μm and tetrodes were moved ~100 μm between sessions. The final tetrode location was confirmed through histological assessment using tetrode tracks.

Offline spike sorting and cluster quality analysis were performed using MClust software (MClust 4.4.07, A.D. Redish) in MATLAB (2015a). Briefly, for each tetrode, single units were isolated by manual cluster identification based on spike features derived from sampled waveforms. Identification of single units through examination of spikes in high-dimensional feature space allowed us to refine the delimitation of identified clusters by examining all possible two-dimensional combinations of selected spike features. We used standard spike features for single unit extraction: peak amplitude, energy (square root of the sum of squares of each point in the waveform, divided by the number of samples in the waveform), valley amplitude, and time. Spike features were derived separately for individual leads. To assess the quality of identified clusters we calculated isolation distance, a standard quantitative metric[95]. Clusters with an isolation distance >6 were deemed single units. Units were clustered blind to interspike interval, and only clusters with few interspike intervals <1 ms were considered for further examination. Furthermore, we excluded the possibility of double-counting neurons by ensuring that the waveforms and response properties sufficiently changed across sessions. If they did not, we conservatively assumed that we recorded twice from the same neuron, and only included data from one session.

Electrophysiological recordings were obtained from 308 SC neurons in 96 behavioral sessions from 10 Gad2-Cre mice (also used in stimulation/behavior experiments). Details of our analyses of the data obtained from our recording experiments are described below. All neural data analyses were performed in MATLAB (2015a/2019a). Neurons recorded during behavior/recording sessions with fewer than 40 trials in either direction or with a choice-epoch firing rate below 2.5 spikes/s for trials in both directions were excluded from all analyses, resulting in 239 neurons in this data set (Supplementary Table 1). In addition, neurons recorded during stimulation/behavior/recording sessions were included in analyses if at least 10 light-on trials were completed in each direction.

**Histology**. Final tetrode location, fiber, and cannula placement, and viral expression was confirmed histologically (Supplementary Fig. 1). Mice were overdosed with an intraperitoneal injection of sodium pentobarbital (100 mg/kg; Sigma Life Science) and transcardially perfused with phosphate-buffered saline (PBS) and 4% paraformaldehyde (PFA) in 0.1 M phosphate buffer (PB). After perfusion, brains were submerged in 4% PFA in 0.1 M PB for 24 h for post-fixation and then cryoprotected for at least 12 h immersion in 30% sucrose in 0.1 M PB. On a freezing microtome, the brain was frozen rapidly with dry ice and embedded in 30% sucrose. Serial coronal sections (50 μm) were cut and stored in 0.1 M PBS. Sections were stained with 435/455 blue fluorescent Nissl (1:200, NeuroTrace; Invitrogen) to identify cytoarchitectural features of the SC and verify tetrode tracks and implant placement. Images of the SC were captured with a ×10 or ×20 objective lens, using a 3I Marianis inverted spinning disc confocal microscope (Zeiss) and 3I Slidebook 6.0 software. Images were adjusted in ImageJ to enhance contrast.

**Statistics and reproducibility**. In general, normality of distributions was tested using the Anderson-Darling test. Data were analyzed using MATLAB (2015a/2019a) and presented primarily as medians or means ± SEM. $p$ values for each comparison are reported in the figure legends, results and/or methods sections. Mice were excluded from analyses based on the misalignment of the optic fiber with ChR2 expression, lack of ChR2 expression, and misplacement of the optetrode outside of the SC (13 mice were excluded from the final data set).

**Behavioral analyses**. The effect of photoactivating GABAergic SC neurons, glutamatergic SC neurons or YFP controls, and of unilateral infusion of muscimol on choice behavior ("stimulation/behavior"; $\beta_{manipulation}$; Fig. 2b, d; Supplementary Figs. 2, 4b) and model behavior (Fig. 6c, e, g–i) was assessed by logistic regression using Python (Sci-Kit Learn package (version 0.21.3)[96]. The logistic function took the form $p = \frac{1}{1+e^{-\eta}}$, where $p$ is the choice made on a given trial (contralateral choice, $p = 0$; ipsilateral choice, $p = 1$), and $\eta = \beta_0 + \beta_{Odor:I} x_{Odor:I} + \beta_{Odor:C} x_{Odor:C} + \beta_{manipulation} x_{manipulation}$, where $\beta_0$ represents overall choice bias, $x_{Odor:I}$ and $x_{Odor:C}$ represent the strength of the odors associated with the ipsilateral and contralateral reward port, respectively, $\beta_{Odor:I}$ and $\beta_{Odor:C}$ represent the influence of the odors on choice, and $\beta_{manipulation}$ represents the influence of manipulation (i.e., light or muscimol). $x_{Odor:I}$ and $x_{Odor:C}$ are calculated as (fraction of odor $- 0.5)/0.5$ and range from 0 to 1. We used separate terms for left and right odors to allow for the possibility that they asymmetrically influenced choice. We set $x_{manipulation} = 0$ on light-off /saline trials and $x_{manipulation} = 1$ on light-on/muscimol trials. Under these calculations, positive $\beta_{manipulation}$ values correspond to ipsilateral influence; for consistency with the choice preference analysis (see below), all $\beta_{manipulation}$

values were subsequently multiplied by −1 so that positive values correspond to contralateral influence[97]. In addition, L2 regularization (C = 1; default parameter value) was applied to all sessions to account for occasional perfect mouse performance. Sessions were included in analyses if at least 15 light-on trials and at least 100 total trials (light-on + light-off) were completed. Statistical significance was determined with a permutation test: we shuffled the values of $x_{manipulation}$ within a session and recalculated $\beta_{manipulation}$, repeated this process 10,000 times to produce a null distribution of $\beta_{manipulation}$ values, and compared the actual $\beta_{manipulation}$ to the null distribution. We tested for significance at $\alpha = 0.05$.

For display, behavioral data from example experimental (Figs. 2a, c, 4d; Supplementary Fig. 4a) and model (Fig. 6b, d, f) sessions were fit with the above logistic function with $\eta = \beta_0 + \beta_{Odor} x_{Odor}$, where $x$ is the proportion of the ipsilateral odor in the mixture, separately for trials with and without manipulation. Our overall results were unchanged when we compared choice biases calculated as $\beta_0 / \beta_{Odor} + 50$ from these separate fits[24,26]. We used $\beta_{manipulation}$ as our primary quantification of choice bias (Fig. 2b, d; Supplementary Figs. 2, 4b) because it requires fewer assumptions than the method comparing separate fits and thus captures choice bias more accurately (and conservatively) for sessions with large effects.

For sessions with a significant $\beta_{manipulation}$ ($p < 0.05$) in the direction of the behavioral effect, we quantified the effects of manipulation on reaction time by examining the duration between the go signal and odor port exit (Supplementary Fig. 3). For trials on which mice exited the odor port before the go signal, we calculated reaction time based on the estimated mean time the go signal could have occurred. Significance for each session was calculated using a two-tailed Mann–Whitney U-test to compare reaction times between light-on trials and light-off trials separately for each direction. Population effects were determined for each direction by using a Wilcoxon signed-rank test to compare the difference in light-on and light-off median reaction times to zero. Only sessions with at least five light-on trials in both directions were included in reaction time analyses, resulting in the exclusion of 2 Gad2-ChR2 sessions and 9 Vglut2-ChR2 sessions.

To test for a dependence of the strength of the contralateral choice bias on the mediolateral location of GABAergic neuron photoactivation (Supplementary Fig. 9), the relative mediolateral placement of the optetrode for each mouse was determined histologically (Supplementary Fig. 1).

**Optogenetic identification of GABAergic neurons.** Before and/or after behavioral sessions in Gad2-Cre mice expressing ChR2, light was delivered via a diode-pumped, solid-state laser (473 nm; Shanghai Laser & Optics Century) at 8 Hz (10 ms on) or, for a small sample (<20 sessions), at 25 Hz (10 ms on) to record extracellular light-induced activity from the same population of neurons that were recorded during behavior ("stimulation/recording"). Isolated units (Supplementary Fig. 5) were identified as GABAergic based on reliable, short-latency responses to light and high waveform correlations between spontaneous and light-evoked action potentials[98–102]. To correct for cells with high tonic firing rates, responses (i.e., action potentials) had to occur within 5 ms of the onset of light and at a significantly higher probability than responses in 5 ms bins outside of light delivery (i.e., baseline responses). To be maximally conservative, we only considered a neuron to be GABAergic if, out of 5000 baseline responses calculated using randomly selected light-off bins, the neuron responded to light more than every baseline response (Supplementary Fig. 5; $p < 0.0002$) and the average light-driven waveform exhibited a correlation with the spontaneous waveform of at least 0.95 (Supplementary Fig. 5; $n = 301$ total neurons, 90 identified as GABAergic). We increased these bins to 7 ms to analyze GABAergic neurons recorded in stimulation/behavior/recording sessions (Fig. 5). Neurons identified as GABAergic could be further identified (i.e., "tracked") during behavior/recording sessions based on spike features and location (i.e., tetrode number and lead number; Supplementary Fig. 5). Neurons from these recordings were also utilized to detect local neuronal responses to light (Supplementary Fig. 6).

**Local neuronal responses to GABAergic activation.** To determine whether rebound excitation was occurring following GABAergic photoactivation during stimulation/recording sessions, the firing rate of each unidentified neuron ($n = 198$) was calculated during the 500 ms preceding light delivery and compared to the firing rate calculated during the 500 ms immediately following the end of the last light pulse in each trial[73]. Rebound excitation was tested using a Wilcoxon signed-rank test to compare the median firing rates of all neurons before and after light delivery (Supplementary Fig. 6).

To assess local inhibitory effects of GABAergic photoactivation, each neuron recorded during stimulation/recording sessions ($n = 301$) was tested for a light-induced decrease in firing rate by comparing the baseline firing rate (calculated over a 1 s window preceding light delivery (intertrial interval)) to the firing rate during photoactivation (entire light delivery period, ~1.2 s) using a one-tailed Wilcoxon signed-rank test. Neurons with a significant light-induced decrease in firing rate ($p < 0.05$; $n = 32$) were z-score normalized to baseline firing rates using 50 ms bins (Supplementary Fig. 6).

The effects of GABAergic photoactivation on local neuronal firing rates (light-excited, light-inhibited and no change; Supplementary Table 2) and absence of rebound excitation[73] (Fig. 3c, d) were verified during behavior by comparing the firing rate of neurons on light-on and light-off trials (separately for ipsilateral and

contralateral choices) within a single session. Neurons were included in analyses if their median firing rate was >0 for the epoch and direction analyzed. For example, a neuron may be excluded from the choice epoch analysis for ipsilateral trials but included in analyses of contralateral trials. To compare light-induced changes between the choice and post-choice epochs (Fig. 3e), the mean firing rate on light-on trials for each neuron was calculated separately for ipsilateral and contralateral choices and for the choice and post-choice epochs. The respective light-off mean was then subtracted from the firing rate in its respective epoch on each light-on trial. The resulting differences were concatenated across the ipsilateral and contralateral directions and the differences were compared between epochs.

Average PSTHs constructed for light-excited and light-inhibited neural subpopulations (Fig. 3f–h; Supplementary Table 2) were z-score normalized to baseline activity (calculated over light-on and light-off trials) using 10 ms bins. A similar method was applied to light-off trials (baseline calculated over ipsilateral and contralateral choices) to construct average PSTHs in Supplementary Fig. 7.

**Waveform analysis.** Standard waveform analyses were performed to attempt to identify GABAergic SC neurons based on distinguishing waveform features[103] (Supplementary Fig. 5). Briefly, the average waveform from each lead was calculated; the lead with the highest peak amplitude was used for analysis. The peak of the waveform was equivalent to the maximum voltage reached during depolarization. Pre- and post-valley were the minimum voltage reached before and after depolarization, respectively.

**Choice preference.** To examine the dependence of the firing rate of individual neurons on choice (Fig. 4c), we used an ROC-based analysis[76] that quantifies the ability of an ideal observer to classify whether a given spike rate during the choice epoch was recorded in one of two conditions (here, during ipsilateral or contralateral choice). We defined the choice epoch as beginning 100 ms after odor valve open and ending with the go signal. We defined choice preference as 2 ($\mathrm{ROC}_{area}$ − 0.5), a measure ranging from −1 to 1, where −1 denotes the strongest possible preference for ipsilateral, 1 denotes the strongest possible preference for contralateral, and 0 denotes no preference[74,75]. Statistical significance was determined with a permutation test: we recalculated the preference after randomly reassigning all firing rates to either of the two groups arbitrarily, repeated this procedure 500 times to obtain a distribution of values, and calculated the fraction of random values exceeding the actual value. We tested for significance at $\alpha = 0.05$.

To examine the timing of choice-related activity of GABAergic neurons with a significant contralateral preference (Fig. 4c, bottom), activity was averaged across contralateral and ipsilateral trials (separately) in 10 ms bins (Fig. 4e). The activity of each neuron was rescaled during the choice epoch from 0 (lowest activity; black) to 1 (highest activity; white) and smoothed with a gaussian filter ($\sigma = 60$ ms). Example peristimulus time histograms in 4a were smoothed with a Gaussian filter ($\sigma = 18$ ms).

Conditional psychometric functions were constructed based on choice-epoch firing rate using the same method described above (Fig. 4d; Behavioral analyses). Firing rates during the choice epoch of GABAergic neurons with a significant contralateral preference (Fig. 4c, bottom) were sorted from lowest to highest. Psychometric functions were then constructed based on choices made on trials in which firing rate was high (top quartile) or low (bottom quartile).

Choice as a function of the number of light-induced action potentials from contralateral-preferring GABAergic neurons recorded during stimulation/behavior/recording sessions (Fig. 5; Supplementary Table 1) was fit to an exponential function, $y = 1 - \alpha e^{-kx}$, where $y$ is the choice on each trial (0 = ipsilateral choice; 1 = contralateral choice), $x$ is the number of light-induced action potentials, $k$ is the slope of the curve (reported free parameter) and $a$ is an additional free parameter to set the intercept.

**Attractor model.** To study how SC circuitry underlies spatial choice, we constructed a rate-based bump attractor model[17,78] consisting of 200 excitatory ($E$) cells and 100 inhibitory ($I$) cells per SC (600 cells total), an E:I ratio consistent with the intermediate and deep layers of the SC[39,40]. Intra-SC synaptic weights were larger for nearby cells, and smaller for more distant ones, determined by

$$W_{ij} = amp * e^{-(i-j)^2 / scale^2}, \quad (1)$$

where $i$ and $j$ are the locations of the pre- and post-synaptic cells, respectively, and amp and scale are defined independently for presynaptic $E$ and $I$ cells (ampE = 0.01, ampI = 0.15, scaleE = 0.048, scaleI = 0.65). amp sets the amplitude (i.e., "strength") of the connection weights, and scale determines the spatial extent over which the connection strength decays. Overall, $I$ cells had a higher scale than $E$ cells, producing conditions for recurrent excitation in the model that are observed experimentally in the SC[104,105]. Inter-SC synaptic connections were made sparse by assigning synaptic weights to a subset of probabilistically determined contralateral $E$ and $I$ cells (InterEamp = 0.003, InterIamp = 0.6, InterEscale = 0.3, InterIscale = 0.6) while all other weights were set to zero. Inter-SC connections were reciprocated between the left and right SC. To promote network stability, each $W$ was normalized to have a maximum eigenvalue of 1.5 by dividing all connection values by max($\lambda$)/1.5, where max($\lambda$) is the largest eigenvalue of the matrix after initialization.

Spike rates of $E$ and $I$ cells in the left SC ($r_E^L$ and $r_I^L$, respectively) evolved at each time step (~2 ms in our numerical simulations) according to

$$\tau \frac{dv_E^L}{dt} = -v_E^L + W_{EE}^L r_E^L - W_{IE}^L r_I^L + W_{R_E \to L_E}^L r_E^R - W_{R_I \to L_E}^L r_I^R + \text{external}^L + \text{noise}$$

$$(2)$$

$$\tau \frac{dv_I^L}{dt} = -v_I^L - W_{II}^L r_I^L + W_{EI}^L r_E^L - W_{R_I \to L_I}^L r_I^R + W_{R_E \to L_I}^L r_E^R + \text{external}^L + \text{noise}$$

and all spike rates were rectified at each time step according to

$$r = \begin{cases} 0, & \text{for } v < 0 \\ V, & \text{for } 0 < v < 100 \\ 100, & \text{for } v > 100 \end{cases} \quad (3)$$

where e.g., $W_{EE}^L$ represents synaptic weights from left $E$ to $E$ cells, $W_{IE}^L$ represents weights from left $I$ to $E$ cells, and $W_{R_I \to L_I}^L$ represents weights from right $I$ to left $E$ cells. Noise was drawn from a Gaussian distribution with mean = 0, and variance = 15 for $E$ cells and 5 for $I$ cells. $r_E^L$ and $r_I^L$ are vectors, with one entry per $E$ or $I$ cell in the left SC, respectively. Similarly, $r_E^R$ and $r_I^R$ describe the firing rates of cells in the right SC, and they evolve over time via the same equations as those in the left SC (i.e., via Eqs. [2, 3], with all "L"s and "R"s swapped).

The vectors external$^L$ and external$^R$ represent the drive to the $E$ and $I$ cells from sources outside the SC. For all trial types, external drive was applied in a linearly graded fashion to the last quarter of cells ($E$ cells numbered 151–200 and $I$ cells numbered 76–100). On leftward trials, the right SC received a stronger external drive than the left SC, and vice versa on rightward trials. Specifically, on "easy" leftward trials, the drive to left SC cell $i$ was external$_i^L = 0.5i + 0.8$, and the drive to right SC cell $i$ was external$_i^R = 9.5i + 0.8$; and on "easy" rightward trials, the drive to left SC cell $i$ was external$_i^L = 9.5i + 0.8$, and the drive to right SC cell $i$ was external$_i^R = 0.5i + 0.8$. For each trial, external drive began at the start of the "choice epoch" (at time step 151), corresponding to the odor delivery time in the task, and stopped at the end of the choice epoch (at time step 400) for an approximate total time of ~500 ms. At step 401, immediately following the choice epoch, the mean $E$ cell firing rate was calculated and choice was determined by the SC with the highest rate. Each "session" consisted of 255 trials, with the proportion of different difficulties matched to the behavioral sessions. Further, to recapitulate mouse-to-mouse experimental variation in viral expression and stimulation efficacy, simulated manipulations occurred at different, randomly selected locations across sessions (although the same cells were manipulated within a single session). To simulate photoactivation of GABAergic neurons, a cluster of 30 $I$ cells were given additional excitatory input during the choice epoch by increasing the rate of $I$ cells by 50 at ~8 Hz (5 model steps on, 60 model steps off). Photoactivation of glutamatergic neurons was simulated by giving additional excitatory input to 30 $E$ cells using the same stimulation strategy above, except instead of increasing the rate by 50, the rate was increased by 20. Decreasing the amount of additional excitatory input was analogous to decreasing the laser power for photoactivation experiments (Fig. 2e, f), a tactic employed to reduce overt movements upon Vglut2-ChR2 SC activation (Methods: Optogenetic photoactivation).

To simulate muscimol inhibition, a cluster of 100 $E$ cells and the neighboring 50 $I$ cells were given inhibitory input throughout the duration of each trial by subtracting 90 from the rate of these neurons at each step. To simulate inter-SC GABAergic terminal activation, $I$ cell terminals were activated on a cluster of 20 $E$ cells and 10 $I$ cells by giving additional excitatory input, proportional to the synaptic weights of the inter-$I$ connections, at ~8 Hz during the choice epoch.

All model results were analyzed using the same methods described for behavioral analyses (Methods: Behavioral analyses).

**Reporting summary**. Further information on research design is available in the Nature Research Reporting Summary linked to this article.

## Data availability
The data that support the findings of this study are available from the corresponding author upon reasonable request.

## Code availability
Custom MATLAB code is available from the corresponding author upon reasonable request.

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

## Acknowledgements

We thank Drs. Joshua Dudman, Jennifer Hoy, Abigail Person, Dan Tollin, John Thompson, and Joel Zylberberg as well as members of the Felsen lab and Matthew Becker for their insightful comments on the manuscript. We thank Nathan D. Baker for technical assistance and Dr. Michael Hall for machining. Light microscopy was performed at the University of Colorado Anschutz Medical Campus Advance Light Microscopy Core and engineering support was provided by the University of Colorado Optogenetics and Neural Engineering Core. Both cores are supported in part by the Rocky Mountain Neurological Disorders Center (P30NS048154), by NIH/NCRR Colorado CTSI grant UL1 RR025780, and by the University of Colorado NeuroTechnology Center. This work was supported by the National Institutes of Health (R01NS079518, F31NS103305).

## Author contributions

J.E. and G.F. designed the experiments. J.E. performed recording and behavioral experiments and analyzed the data. J.B.H. performed behavior experiments and analyzed behavioral data. J.E. constructed the model and analyzed the results. J.E. and G.F. guided data analysis and oversaw the project. All authors discussed the results and J.E. and G.F. wrote the manuscript.

## Competing interests

The authors declare no competing interests.
