## [Peer Review File · Communications Biology]

This manuscript has been previously reviewed at another Nature Research journal. This document only contains reviewer comments and rebuttal letters for versions considered at Communications Biology

Reviewers' comments:

Reviewer #2 (Remarks to the Author):

This review is a resubmission of a paper that I previously reviewed for another journal. The manuscript has improved in that some of the statements that aren't fully backed up by the data have been softened a bit.

I do have remaining concerns, however, related to my previous comments:

- In the response to reviewers, the authors included the GABAergic responses encompassing the choice and post-choice epochs, showing that the choice-selective neurons active during the choice epoch are a minority. The strongest GABAergic activity seems to actually be in the post-choice epoch. However, the photoactivation experiment is triggering activity in the entire GABAergic population during the choice epoch. It is promising that Fig 4d shows that the photoactivation effects on behavior are stronger on trials when the choice selective subpopulation of GABAergic cells are more active. Even this result is not conclusive though, as this trial-trial variability could also be correlated across the entire GABAergic population.

This experiment doesn't demonstrate conclusively that long-range inhibition out of the SC normally occurs during the choice period, rather it shows that when the full GABAergic population is activated in this way, long-range effects seem to dominate over local effects. This issue needs to be acknowledged and addressed.

- The axon terminal stimulation experiment is still included, but moved to supplementary. I appreciate that it wasn't just removed, but now it's described in the discussion as if it bolsters the claims of the paper. ("Activation of GABAergic terminals 580 resulted in a similar contralateral choice bias as somatic stimulation (Supplementary Fig. 581 7c,d), although in this preliminary data set, the choice bias was significant only in individual 582 sessions ($p < 0.05$; 3/13 sessions) and not for the population ($p = 0.6355$, $W = 53$, two-tailed Wilcoxon signed-rank test). ") It does not if there is only an effect in 3/13 sessions. Instead, it possibly provides evidence that is it not simply the intercollicular GABAergic projection that is responsible for the behavioral effect, but that that GABAergic collaterals to other regions may also affect choice bias. As pointed out by the other reviewer, the experiments in this paper do not demonstrate that the long-range inhibition specifically to the other SC is responsible for the effects of unilateral GABAergic stimulation. The claims made by the paper should be made more general, or additional experiments done to demonstrate specifically where the GABAergic projections terminate, and their direct impacts on the other SC.

Reviewer #3 (Remarks to the Author):

Essig et al. Communications Biology 2020

Karel Svoboda

Generally speaking this is a very interesting paper on SC in left/right choice behaviors, with many intriguing results. The experiments and modeling represent a lot of hard work and are competently executed. These studies will be the basis of future studies, which is more than one can say about the vast majority of systems neuroscience experiments. The paper deserves to be published.

Below we make comments to help clarify some points and perhaps prevent the authors of painting themselves into a corner, because we believe a lot remains to be discovered about the roles of SC in

these simple choice behaviors.

Major comments:

Perhaps the most surprising finding to us is that SC GABAergic neurons can be activated without a decrease in excitatory activity (Figure 3). This means that the SC is not an inhibition stabilized network (see Li, Chen et al 2019 eLife; Sanzeni et al 2020). Yet they model SC as an inhibition stabilized network. There is a contradiction here. We suspect that unknown multi-regional interactions play a major role.

The obvious experiments of activating GABAergic neuron terminals to elicit ipsilateral biases (i.e. wrt to photostimulation) did not work (sFig 7). This is surprising and in some sense falsifies hypothesis 2 (if done properly -- probably 10x more power required for axons; see Huber et al 2008).

Fig. 3b - Upon photoactivation of GABAergic neurons there seems to be a change also in activity of "ipsi-unidentified" neurons. The size of this effect appears to be of the same order as the increase for ipsi-GABA, but according to the author's statistics - completely not significant. As the authors mention, this result has an important implication to the hypothesis that GABA-ergic activation primarily affects the neural activity on the contra-lateral side. Therefore, the effect of GABAergic activation on the unidentified cells (presumably glutamatergic cells) should be quantified in much greater detail and shown on the level of individual cells as well as population-averaged PSTHs. Are those cells that are inhibited (Supplementary Figure 4) by GABAergic photoactivation are special in any way in terms of their PSTH-response profile? Are those cells that are excited (non-directly) by GABAergic photoactivation have particular task-related response profiles?

Supplementary Table 2 is not very informative and should be replaced by a figure where population-averaged PSTHs of the corresponding groups of neurons are shown, in order to assess the magnitudes and the timescales of the effects.

I don't suggest new experiments, but in case the authors already have this data - it would be nice to see if indeed ipsi-lateral activation of GABA ergic neurons inhibits the activity of neurons on the contra-lateral SC.

The authors suggest that ipsi-lateral GABAergic inactivation has no effect on the activity of ipsi-lateral principal cells, but instead might affect the contra-lateral SC. This might be inconsistent with long-range inhibition hypothesis: According to this hypothesis, one may expect that if the activity of the contra-lateral neurons is biased by ipsi-lateral GABAergic photoactivation, they should in turn inhibit the SC neurons (via inhibitory neurons projecting from the contra to the ipsi side). The authors might want to explore this point using their network-model.

Minor comments:

The authors might compare and contrast bilateral models of ALM (Li/Daie et al 2016; Inagaki et al 2020) and keep in mind that the SCm is bidirectionally connected to frontal cortex.

Introduction: Most research about the role of SC in spatial cognition was conducted in primates. It would help the reader if the authors specify which papers that are mentioned in the introduction are based on SC studies in rodents.

Page 4 (line 85): The functional outcomes of the two hypotheses are not very clear, especially for the second hypothesis. They become more clear when the authors refer to Figure 1, but these happens only at the very end of the introduction

Figure 1b,d - what is represented by triangles, circles? The logic why some of the lines are in bold is not immediately clear.

It is unclear how the hypotheses about local versus long-range inhibition map onto winner-take-all/race-to-threshold models mentioned in the introduction and the discussion. The authors mention in the discussion that race-to-threshold models "do not require such inhibitory interactions", but it would help the reader to know that earlier - in the introduction, when the two alternatives are described.

Does the effect of photostimulation of GABAergic neurons on reaction time consistent more with race-to-threshold or winner-take-all models?

Fig. 2c. The effect of ipsilateral muscimol injection (ipsi-lateral bias) can be in principle attributed to

an increase in the effect of long-range inhibitory projections from the contra-lateral side. The authors should discuss this possibility.

The result in Fig.2e is almost trivial, and does not advance the paper, which is focused on GABAergic neurons. I suggest moving it to the supplementary, and discussing more the difference between Fig.2a and Fig.2e.

Page 11 (line 254): "GABAergic firing rates" should probably be "GABAergic neurons firing rates". What do the authors refer to as "ipsi" "contra" choices in Fig.3b-c? Does it refer to the selectivity of the neurons?

The authors should show more example neurons in Fig. 4 and also the population average PSTH for GABAergic and non GABAergic neurons.

What is the interpretation of the result presented in Fig.4d?

The authors should discuss if the fact that the GABAergic activity is higher on contra-lateral trials (Figure 4) may be because local GABAergic neurons may reflect the population average activity of local principal neurons (which increase their activity on contra-lateral trials). This can be also explored using the attractor model that the authors present in this paper.

Discussion (page 22, 533) - what do the authors refer to as "multiple SC loci"? I found the entire discussion (page 22-23) regarding "winner-take-all" versus "winner-take-most" mechanisms and "SC loci" totally unclear. Specifically, it was unclear what is the difference between these two mechanisms and how does the current data distinguish between them.

Discussion (Page 23): the fact that most GABAergic neurons were indifferent to choice, does not necessarily mean that they are not part of the choice-selection mechanism (see Najafi et al Neuron, 2020 "Excitatory and Inhibitory Subnetworks Are Equally Selective during Decision-Making and Emerge Simultaneously during Learning".)

Reviewer #4 (Remarks to the Author):

In this study, Essig et al aim to investigate the role of GABAergic neurons in SCid in spatial decisionmaking. This question is an important one, and the authors have expended significant effort in attempting to address this question: they use a two alternative forced choice behavior combined with cell-type specific optogenetic manipulation, as well as opto-tagged recordings; kudos on the effort. The authors activate GABAergic neurons in the intermediate and deep SC (SCid) layers and seem to observe a, counter-intuitive, contraversive shift in choice bias. (Past literature has identified GABAergic

neurons in the SCid to have both intracollicular as well as cross-collicular projections.) To support their observation the authors make the claim that photoactivating GABAergic neurons mainly affects crosscollicular projections. Further, the authors use optotagged recordings from GABAergic neurons to show

that a subset of the tagged GABAergic neurons show activity that correlates with contraversive choice. Finally, the authors attempt to support their claim of GABAergic inhibition working via the crosscollicular projections by constructing an attractor model.

However, the results and the experimental techniques used in the study do not support the claims and the interpretation of the results that the authors lay out in the manuscript. This discrepancy between results and interpretation significantly impacts and narrows the scope of the results. A combination of potentially improperly analyzed data, ambiguous (and not clearly interpretable) results, sub-par writing,

and overextended claims in terms of the circuitry as well as modeling make the paper unsuitable for publication in its current form. (Some of these concerns appear to have already been raised by previous reviewers, and the authors do not seem to have addressed them satisfactorily.) Improved writing and scholarship, better analysis, significantly reduced claims re theoretical/modeling issues, coherent flow, and a tight presentation of a story focused primarily on the experimental questions and

results – all well within the capability of the authors - will address many of these concerns. However, even with that done, the fundamental scientific issue at the very heart of the study will still remain unresolved/uninterpretable from the data presented, which is what significantly dampens enthusiasm for the paper, unfortunately. Details below.

Detailed comments/The story and findings.

The clearest story that appears to be extractable from the results is as follows. (This required multiple reads of the difficult-to-follow manuscript.) Concerns are highlighted in blue.

1. The role of SCid GABAergic neurons in spatial decision-making is not clear (although see very recent study by Wang...Krauzlis 2020, J Neurosci). Based on the wealth of literature about the SCid including results from manipulating excitatory neurons in SCid with electrical stimulation and muscimol delivery, results from slice experiments as well as our knowledge of SCid anatomy, a traditional expectation would be that activating GABAergic neurons in one hemisphere would suppress nearby excitatory neurons, resulting in suppression of contralateral behavioral choices/responses.

2. In order to test the role of SCid GABAergic neurons, the authors stimulate GABAergic neurons in GAD-cre mice. [This is already problematic for two reasons: 1) Activation experiments only reveal sufficiency, not the "role" of the neurons in normal behavioral function, 2) Specifically, activation of GABAergic

neurons within circuits in the cortex as well as in SCid is an experimental "trick" used to shut down nearby

excitatory neurons as opposed to investigate the "role" of the GABAergic neurons themselves. Not only do the

authors not acknowledge this, they have on more than one occasion, misinterpreted the

Wang...Krauzlis

2020 paper by referring to the results as showing the function of GABAergic neurons, whereas the original

authors of the paper state clearly that the results are better viewed as the consequence of silencing excitatory

neurons. Nonetheless, activating GABAergic neurons and examining the effect of this manipulation on neural

responses and behavior is a fine thing to do. So let us proceed.]

Contrary to expectation, they potentially find that activation of GABAergic neurons promotes contralateral responses. [I say 'potentially', because here, as in most figures, the unit of statistical analysis,

i.e., n , that they use is a session. This is not appropriate, and is also not the convention. The appropriate n

should be number of animals (and this should be specified clearly within each figure legend, as opposed to

once in the Methods). The reason this issue is critical is: say, the number of animals used in Figure 2a (the

lynchpin result of the paper) was 8, with an uneven number of sessions per animal. It is possible that the

result was robust in one or two animals and was not apparent in the others. If, out of the total number of

sessions reported, a disproportionately large fraction of them was from these two animals, this could likely

result in their finding being not true, statistically. Unless the data are analyzed and plotted in terms of $n=8$

animals (rather than the current $n=96$ sessions), it is impossible to assess the validity of the finding. This

analysis is easy to re-do, and for the purposes of this review, let us proceed with the assumption that the

finding will still stand true once the analysis and plotting are re-done correctly – i.e., GABAergic neuronal

activation promotes contralateral choices.]

3. As a sanity check in the face of this unexpected/puzzling result, the authors do two experiments: silencing excitatory neurons pharmacologically, and activating excitatory neurons optogenetically (in VGLUT-cre mice), and they find the expected result in both cases. [This was not really necessary to do, in my opinion, and could easily be moved to supplementary data, because the muscimol experiments and

the scientific outcome of the optogenetic activation of excitatory neurons are very well established.]

4. To investigate this central puzzling result (Fig. 2A), the authors perform opto-tagged recordings. They verify this approach in Figure 3, show in Figure 4 that a greater fraction of opto-tagged GABAergic neurons show responses correlated with contralateral choice than ipsilateral choice, and show in Figure 5 that greater activity of opto-tagged GABAergic neurons correlates with higher likelihood of contralateral choice. [I appreciated their demonstration in Fig. S3 that in the SCid, attempting to classify GABAergic neurons simply on the basis of waveform shape is not effective]. [For clarity, these figures could be grouped into one or at most two figures, with the rest of the info moved to

supplementary. Importantly, the motivation for and description of these results can be described more succinctly and clearly]

5. The authors then hypothesize that one account for this unexpected finding could be greater strength of intercollicular inhibition compared to intra-collicular inhibition (Inter-collicular inhibition is known, anatomically, to exist, as they point out in the intro). They illustrate this idea using a computational model (Figure 5). [This is the only utility I see of the model as it has been presented – as a

quantitative illustration of this idea of greater intercollicular versus intracollicular inhibition efficacy. This is

essentially a sophisticated curve-fitting exercise, in the sense that a model has been built to fit/explain their

data. However, this is only one half of the utility of any good modeling effort because one can always fit a

“curve” (model) to data. The true worth of a model is measured by the testable predictions it makes, and the

testing of the predictions. Since the authors do not do this, the model serves only to account for their data,

meaning that any number of alternative accounts may also be valid. Further, their setting up of two ‘competing’ modeling perspectives is largely a straw-man argument. Although the race models (or accumulation to bound models) proposed originally did not include any inhibition (and involved independent accumulators), modern views of these models includes inhibition, and has done so for at least 10 years now.

There are many review papers, particularly on perceptual decision-making that cover this ground (for instance, Churchland et al, 2012, Curr Opin Neurobiol; Mysore et al 2020, eLife). In this context, the fact that

the authors heavily infuse ideas about models of decision-making throughout their paper, comes across as

not appropriate. In fact, the entire set-up of the paper, right at the outset of the introduction, is that the study is

going to help us understand models of decision-making better, which is disingenuous at best- I strongly

recommend that the authors simply remove lines 37-49, and begin the intro at “Spatial choice-...”, which

would allow the paper to be focused on the experimental question and results. (Potentially, the narrative laid

out in the points here could be one that the authors may consider using.) They would also need to significantly

streamline the Intro to keep this focus on the experimental question. Any mention of modeling should

be done

in a limited manner in the Discussion.]

6. The next obvious step would be to test experimentally, this hypothesis. The way to do this would be to activate cross-hemispherically projecting GABAergic fibers to test if this can promote contralateral choices. This is the critical experiment to test their hypothesis. [I originally thought that they had not done this experiment and was puzzled by why they hadn't. It turns out that they actually did do

this experiment, but the results are only found in Supplementary Figure 7 and seem to only be brought up in

the Discussion! It turns out that the results do not appear to support their hypothesis. Out of 11 sessions, only

3 show the expected result. This is problematic for a few reasons: 1) As in previous figures, this should be

analyzed and plotted in terms of number of animals and not number of sessions. How many animals were

tested here? What is the result if data are plotted with n =animals? 2) If their hypothesis is true, most sessions

should show an effect in the predicted direction, which does not seem to be the case. Therefore, this critical

experiment seems to not support their hypothesis for why activating GABAergic neurons would promote

contralateral choices, leaving us to consider alternative explanations (more on this in point #8). As a side note,

this very issue was brought up also by a previous reviewer. The authors' response seems to be that this is

'preliminary' data and to move it to supplementary, neither of which action actually responds to the question!

The notion of 'preliminary' data in a paper is not meaningful. And moving it to supplementary and seemingly

brushing it out of view is also not appropriate.]

7. At this point, we are left with the following scientific results: That activation of inhibitory SCid neurons

can cause promotion of contralateral choices, that activity of many GABAergic neurons is correlated with

contralateral choice, that this could be because of stronger inter-collicular inhibition compared to intracollicular inhibition, but that upon testing directly (with fiber activation), this does not seem to be the

explanation. Because the hypothesis does not seem to be supported, the core findings then are simply the

first two, which boil down to "activation of GABAergic neurons can promote contralateral choices, contrary to

traditional expectation". As I said at the outset, even if all the issues with analysis and writing are addressed,

this remains seemingly the primary scientific advance of the study, which is incremental at best. Given that we

know that cross-hemispherical GABAergic projections exist, that they can promote contralateral choice is not

surprising. And I would like to point out again that this has really nothing to do with two different 'opposing'

models, or about achieving deep insights into circuits of models for decision-making. This is simply an unexpected experimental finding.

8. In this context, then, an important point to discuss would be why they find this unexpected result, whereas

others (specifically Wang ... Krauzlis 2020) have found the expected result? This discussion becomes

extremely important if the per animal re-analysis of Figure 2 pans out. For instance, could the difference in results simply be the result of chance differences in the placement of the virus and the proportion of ipsi versus contra projecting GABAergic neurons that are impacted? Could there be differences in the subtypes of GABAergic neurons recruited by using GAD-cre mice vs. VGAT-cre mice (as in Wang et al), coupled with differences in the projection patterns (local versus cross-hemispherical) of subtypes? These possibilities could account for all of their results (Fig.2, Fig 5, Supp Fig 7), especially considering that the results in these figures do not overwhelmingly support their claim, and that there are many instances of the expected result as well. As the data in this paper do not strongly support one (their) explanation over others, it is difficult to conclude definitively. As things stand, then, the best case that can be made from these data, it seems, is that it is possible for GABAergic neurons to promote contralateral choices. This is, unfortunately, not a significant advance nor new or unexpected.

9. As asides, the title is also misleading; their use of "long-range" inhibition should be replaced by intracollicular inhibition throughout because the former phrase is misleading/ambiguous. The title could at best read "Inter-collicular inhibition can mediate goal-directed decision making".

10. Others: There are many other issues with writing, analysis, and scholarship. Only a few are listed here and below because the scientific and structural concerns are more imposing: a) Figure legends are not clear. They should be clear descriptions of the panels as opposed to discussion of the meaning; legends should include stats. b) The need for a GLM type quantification of results in Figure 2 (with the poorly explained betas) was unclear. Similar work in monkeys simply fit sigmoids and quantify the half-max or threshold to assess leftward rightward shifts. c) the discussion does not include treatment of the central issue, described above in point #8. d) Supp Fig 3 is very jumbled; panels need to be reorganized. Additional comments:

1. Figure 1a: Assumption that photoactivation of GABAergic neurons mainly has an effect of inactivating the inter-SC projecting neurons is unsubstantiated.

. Lines 67-70: The authors claim that traditional methods are unable to isolate GABAergic neurons to dissociate contributions of intrinsic and extrinsic SC inhibitory mechanisms. - However, the techniques that the authors use is basically to infect "all" SC GABAergic neurons, and also does not distinguish between inter-collicular and intra-collicular inhibition. (See Wang Krauzlis, 2020)

a. Lines 75-76 - It seems that the authors have misinterpreted a recent study (Wang Krauzlis, 2020) which uses similar methods of activating GABAergic neurons in the SC. Since the techniques are the same, why do the authors interpret their photoactivation as inter-collicular whereas the Wang Krauzlis, 2020 is interpreted as intra-collicular?

b. Lines 102-103: The authors provide no experimental (anatomical) evidence that local GABAergic neurons exert a greater inter-collicular inhibitory effect rather than intra-collicular inhibitory effect.

i. One way could be to reconstruct the terminal fields of the inhibitory neurons

ii. The other way could be to inhibit activity of inter-collicular GABAergic projections during behavior. This would give a direct answer to the role of inter-collicular GABAergic projections in selection.

c. Supplementary Figure 7: The results of this data are inconsistent and do not support the results of Figure 2a. It seems that the weak +ve correlation is driven mainly by 1-2 behavioral sessions. Please also see previous comments on the need for re-analyzing and re-plotting several figures with comparisons one across mice as opposed to across sessions.

2. Figure 1b: Assumption that using muscimol has mainly a local GABAergic effect is also unsubstantiated (lines 180-181).

a. Muscimol works by activating post-synaptic GABAergic receptors and not via activating GABAergic neurons. Thus, delivering muscimol basically reduces activity of all cells with active GABAergic receptors irrespective of whether these receptors form synapses with local intra-collicular interneurons or with inter-collicular projections or inhibitory projections from outside the SCid.

b. Activating GABAergic post-synaptic receptors or activating pre-synaptic GABAergic neurons (using light) to turn-off upstream excitatory neurons is a common technique used across the field. In these studies, very rarely are comments regarding the functional role of GABAergic neurons made.

We thank the Reviewers and the Editor for their constructive comments. These comments, along with the Editor's specific recommendations, have prompted us to significantly overhaul the manuscript by reframing it to focus on our key experimental findings and by performing several new analyses on our behavioral and neural data. We have addressed all concerns. In the responses below we indicate the pages and line numbers corresponding to significant changes in the "red-lined" version of the manuscript, in which we note significant deletions with crossed-out text and additions with red underlined text.

Reviewer #2

This review is a resubmission of a paper that I previously reviewed for another journal. The manuscript has improved in that some of the statements that aren't fully backed up by the data have been softened a bit.

I do have remaining concerns, however, related to my previous comments:

1. In the response to reviewers, the authors included the GABAergic responses encompassing the choice and post-choice epochs, showing that the choice-selective neurons active during the choice epoch are a minority. The strongest GABAergic activity seems to actually be in the post-choice epoch. However, the photoactivation experiment is triggering activity in the entire GABAergic population during the choice epoch.

We thank the reviewer for continuing to provide constructive feedback on our manuscript. We have appended a response to a concern raised by this reviewer about a previous version of the manuscript (see *, below), in case it applies to the revised version as well.

The Reviewer Figure in the previous response to reviewers addressed the question "The analyses in Figure 4 are quite convincing – how does inhibitory activity unfold over the entire trial, including after the go signal?" This figure showed that many GABAergic SC neurons were active following the go signal, as mice leave the odor port and orient to the reward port. As is the case in our unidentified population and in other studies of the SC (Munoz and Wurtz, 1995; Basso and Wurtz, 1998; Everling *et al.*, 1999; Felsen and Mainen, 2008), SC neurons often exhibit more activity during movement than during choice. We are quite interested in this movement-related activity and its potential role in motor control; this is the focus of a separate manuscript in preparation. However, given our current focus on spatial choice, we focus on the activity of GABAergic SC neurons during target selection, and not during movement after the choice has already been made. We agree that many of the identified GABAergic SC neurons do not exhibit choice preference during the choice epoch (represented by the white bars in the bottom panel of Fig. 4c) yet are part of the population being photoactivated in our stimulation/behavior sessions (Revised Fig. 2a,b) (see response to Comment 3). These neurons may play a role in spatial choice that was not measured by our experimental paradigm, or they may contribute only to other SC functions (see response to Comment 28). Nevertheless, our finding that activating the entire population of GABAergic SC neurons during choice promotes the selection of contralateral targets is consistent with our finding that more neurons in this population exhibit contralateral than ipsilateral choice preference (Fig. 4c, bottom, 25 neurons prefer contra, 4 prefer ipsi ($p = 9.7 \times 10^{-5}$, X^2 test) (p. 16, lines 378–380). We would expect that activating GABAergic neurons that do not exhibit a preference during the choice epoch (Fig. 4c, bottom, white bars) would not promote the selection of either choice, but would only add noise to the results (also see response to Comment 3).

2. It is promising that Fig 4d shows that the photoactivation effects on behavior are stronger on trials when the choice selective subpopulation of GABAergic cells are more active.

We have clarified in the Revised Fig. 4 legend that data in Fig. 4 are from behavior/recording sessions, which do not include photoactivation (p. 16, line 387). Fig. 4d shows that mice are more likely to select the contralateral port on those trials in which the endogenous activity of contralaterally-preferring GABAergic SC neurons is higher during the choice epoch. Instead, the reviewer appears to be describing the relationship we observe in Fig. 5 which shows data from stimulation/behavior/recording sessions demonstrating that contralateral choice is more likely on light-on trials in which more spikes were elicited by light.

3. *Even this result is not conclusive though, as this trial-trial variability could also be correlated across the entire GABAergic population.*

Our understanding of the reviewer's overall concern (which we have split into Comments 1-3 to most clearly address) is that, in Revised Fig. 2a,b, photoactivation was not restricted to GABAergic SC neurons that exhibited a choice preference during the choice epoch (Fig. 4c, bottom, black bars), and we have therefore not conclusively shown that the activity of these neurons promotes contralateral choices. We agree that photoactivation was not restricted to a subset of GABAergic SC neurons. If it were possible to restrict photoactivation to this specific population of GABAergic SC neurons we would expect to observe a stronger effect on choice; the unavoidable effect of photoactivating – directly or indirectly – other populations of neurons that are not involved in choice would only add noise. We have acknowledged at several places in the revised manuscript that GABAergic SC neurons are a heterogeneous and diverse population and that our photoactivation experiments affected the activity of the entire population (p. 2, lines 30–34; p. 4, lines 84–86; p. 5, lines 109–111; p. 5, lines 119–122; p. 6, lines 132–135; p. 7, lines 149–151; p. 10, lines 237–240; p. 16, lines 378–380; p. 21, lines 525–527; p. 26, lines 646–648). As genetic markers for subtypes of GABAergic SC neurons are identified, it will become possible to perform more restricted photoactivation experiments (p. 26, lines 648–650).

4. *This experiment doesn't demonstrate conclusively that long-range inhibition out of the SC normally occurs during the choice period, rather it shows that when the full GABAergic population is activated in this way, long-range effects seem to dominate over local effects. This issue needs to be acknowledged and addressed.*

We agree with the reviewer and have acknowledged in the revised manuscript that the experiments in which we photoactivated GABAergic SC neurons during behavior (Revised Fig. 2a,b) alone “do not offer insight into normally-occurring GABAergic SC activity during spatial choice” (p. 14, lines 348–349). To address this issue, we clarify that “We therefore examined the endogenous activity of these neurons during spatial choice, in the absence of photoactivation” (p. 14, lines 349–351). These recordings demonstrate that the endogenous activity of a subpopulation of GABAergic SC neurons increases during contralateral choice (Figs. 4; 5), “consistent with the contralateral bias elicited by activating the entire population of GABAergic SC neurons during spatial choice” (p. 16, lines 378–380).

While the results of our manipulation experiments (Revised Fig. 2) and stimulation/behavior/recording experiments (Revised Fig. 3) are not consistent with the idea that GABAergic SC neurons suppress local premotor output during spatial choice, we agree that we have not elucidated a complete mechanism for how GABAergic SC neurons mediate spatial choice. One possibility, supported by our model, is that the target of this inhibition is the opposite SC (Fig. 6h,i). But additional experiments will be required to conclusively identify these target/s, which will be the focus of future studies (p. 23–24, lines 576–579; also see response to Comment 5).

5. *The axon terminal stimulation experiment is still included, but moved to supplementary. I appreciate that it wasn't just removed, but now it's described in the discussion as if it bolsters the claims of the paper. (“Activation of GABAergic terminals 580 resulted in a similar contralateral choice bias as somatic stimulation (Supplementary Fig. 581 7c,d), although in this preliminary data set, the choice bias was significant only in individual 582 sessions ($p < 0.05$; 3/13 sessions) and not for the population ($p = 0.6355$, $W = 53$, two-tailed Wilcoxon signed-rank test).”) It does not if there is only an effect in 3/13 sessions. Instead, it possibly provides evidence that it is not simply the intercollicular GABAergic projection that is responsible for the behavioral effect, but that that GABAergic collaterals to other regions may also affect choice bias. As pointed out by the other reviewer, the experiments in this paper do not demonstrate that the long-range inhibition specifically to the other SC is responsible for the effects of unilateral GABAergic stimulation. The claims made by the paper should be made more general, or additional experiments done to demonstrate specifically where the GABAergic projections terminate, and their direct impacts on the other SC.*

Based on the totality of the Reviewers' and Editor's comments, we have removed the results of our terminal stimulation experiments from the revised version of the manuscript to focus on our main experimental finding that GABAergic SC neurons do not locally suppress premotor output. Although we considered our terminal stimulation results to provide preliminary support for the predictions of our attractor model – in particular, photoactivating terminals had a very clear effect on reaction times, indicating their involvement in this spatial choice task – as noted

by Reviewer 3 (Comment 7), we likely provided insufficient light power to fully photoactivate GABAergic SC terminals in the opposite SC. We agree that we have not demonstrated that long-range inhibition to the opposite SC is responsible for the effects of unilateral GABAergic stimulation, and have made these claims more general in the revised manuscript (Revised Fig. 7; p. 24, lines 584–587), including changing the title. Thus, instead of presenting these preliminary data here, we will build on our current experimental and modeling results to focus on the functional role of intercollicular neurons (both GABAergic and glutamatergic) in a future study.

** [Previous comment] Although in their reply the authors say that they have clarified their rationale for assuming local inhibition, the point remains that there has been evidence in the literature for crossed inhibitory projections for some time, so their initial hypothesis remains an odd straw man. And despite the reply, the actual changes to the manuscript on this point are minimal. For example, the abstract still frames this as a viable hypothesis to be rejected.*

In response to the reviewer's concern with a previous version of the manuscript, in the immediately previous submission we framed our experiments as testing two hypotheses: that GABAergic SC neurons most strongly provide local inhibition or intercollicular inhibition. Most notably, we added panels to Original Fig. 1 (panels d,e) showing the predicted effect on spatial choice of photoactivating intercollicular GABAergic neurons under these two hypotheses. In the revised manuscript, to focus more narrowly on our experimental results demonstrating that GABAergic SC neurons do not locally suppress premotor output during spatial choice and to reframe the manuscript to improve readability, as suggested by Reviewer 4, we have removed these new panels and overhauled much of our text. Given that we removed the terminal stimulation experiments (see response to Comment 5), we believe that the readability of the manuscript is greatly improved by framing the manuscript as testing for local suppression of premotor output (see response to Comment 29).

However, we believe these revisions also continue to address the reviewer's previous concern that our hypothesis of local suppression of premotor output neurons is a "straw man." Specifically, a) As suggested by Reviewer 4, we describe how the recent results of Wang et al. (2020), and other studies using photoactivation of GABAergic neurons to produce local inhibition, support the premise for our hypothesis of local suppression of premotor output (p. 5, lines 98–102), and that our results are "unexpected" in light of these previous studies (p. 6–7, lines 144–149; p. 9–10, lines 223–226); b) we explain how previous observations of ipsilateral-preferring SC neurons further supports this premise (p. 3, lines 66–69; p. 5, lines 104–107); c) we continue to describe the known local and long-range projection patterns of GABAergic SC neurons (p. 4, lines 84–97) and what the predicted effect of photoactivation would be if they played balanced roles (p. 5, lines 109–111; p. 7, lines 149–151); and d) we continue to emphasize that, although the anatomy of GABAergic SC neurons is known (Mize, 1992; Behan *et al.*, 2002; May, 2006; Sooksawate *et al.*, 2011), to understand their role in spatial choice their activity must be recorded and manipulated in behaving animals (p. 4, lines 95–97).

Reviewer #3 (Karel Svoboda)

Generally speaking this is a very interesting paper on SC in left/right choice behaviors, with many intriguing results. The experiments and modeling represent a lot of hard work and are competently executed. These studies will be the basis of future studies, which is more than one can say about the vast majority of systems neuroscience experiments. The paper deserves to be published. Below we make comments to help clarify some points and perhaps prevent the authors of painting themselves into a corner, because we believe a lot remains to be discovered about the roles of SC in these simple choice behaviors.

We thank the reviewer for his constructive comments. We share the belief that much remains unknown about how the SC mediates spatial choice and we look forward to making further progress by building on the results of this study.

Major comments:

6. Perhaps the most surprising finding to us is that SC GABAergic neurons can be activated without a decrease in excitatory activity (Figure 3). This means that the SC is not an inhibition stabilized network (see Li, Chen et al 2019

eLife; Sanzeni et al 2020). Yet they model SC as an inhibition stabilized network. There is a contradiction here. We suspect that unknown multi-regional interactions play a major role.

This is a very interesting point that requires some unpacking. First, because we were conservative in identifying GABAergic neurons with our optotagging approach (Supp. Fig. 5e; p. 15, lines 354–359), our unidentified population of neurons likely includes some GABAergic neurons, that either a) didn't meet our stringent criteria to be identified as GABAergic or b) were not expressing ChR2. We are therefore cautious about making claims about the activity of excitatory neurons based on the results of our stimulation/behavior/recording experiments (Revised Fig. 3).

Second, we did not design our experiments to test whether the deep layers of the SC are organized as an inhibition stabilized network (ISN), and we would be hesitant to conclude that it is or is not based on our stimulation/recording data. Because we were familiar with the method of producing inhibition by activating inhibitory interneurons (i.e., the photoinhibition “trick” noted by Reviewer 4 (Comment 31)), as well as the paradoxical effects that can be observed with this approach (Li et al., 2019), we performed stimulation/behavior/recording experiments to verify that, during spatial choice, light delivery at the same power used for our manipulation experiments (Revised Fig. 2a,b) had the desired effect of transiently increasing the activity of GABAergic neurons (Fig. 3). However, we did not perform the array of experiments required to determine whether this circuit functions as an ISN.

Third, we do not claim that activating GABAergic SC neurons does not produce any decrease in excitatory activity. On the contrary, a subpopulation of unidentified neurons (which may or may not be excitatory), exhibited a light-induced decrease in activity during the choice epoch (11/51 neurons, Supp. Table 2; Revised Fig. 3h). These results support the idea that GABAergic SC neurons provide local inhibition (Revised Fig. 7). However, in the revised manuscript we show that the neurons that are inhibited are unlikely to be premotor output neurons: as shown by their average PSTHs, they do not exhibit the contralateral choice preference typical of premotor output neurons (New Supplementary Fig. 7b; see response to Comment 10; p. 17, lines 411–416).

We thus suggest that it is not contradictory to include local inhibition, which is indeed required to stabilize the network, in our attractor model. Similar networks have been used in other studies of SC function (Kopecz, 1995; Trappenberg et al., 2001; Lo and Wang, 2006; Kopec et al., 2015), and we believe the model is useful for recapitulating our results and for making testable predictions, including that the contralateral choice bias observed upon activating a heterogeneous population of local and long-range GABAergic SC neurons (Revised Fig. 2a,b) is due to their relatively strong inhibition of premotor neurons in the opposite SC (Fig. 6h,i).

Nevertheless, we entirely agree that multi-regional interactions play a role in spatial choice, and possibly even in how the SC mediates spatial choice (p. 24, lines 581–584). In the revised manuscript, we have clarified that the intent of our model is to examine how our results could be explained without invoking extracollicular structures (p. 24, lines 584–587), and we have discussed our results in the context of studies on ALM circuits (p. 24, lines 587–598; see response to Comment 14).

7. The obvious experiments of activating GABAergic neuron terminals to elicit ipsilateral biases (i.e. wrt to photostimulation) did not work (sFig 7). This is surprising and in some sense falsifies hypothesis 2 (if done properly -- probably 10x more power required for axons; see Huber et al 2008).

We thank the reviewer for pointing out that we likely used insufficient light power to effectively activate the terminals of GABAergic neurons. Based on this feedback as well as on other comments of the reviewers and Editor, we have removed our terminal stimulation experiments. We intend to build on the current study to focus on the functional role of intercollicular neurons (both GABAergic and glutamatergic) in future studies (p. 23–24, lines 576–579).

8. Fig. 3b - Upon photoactivation of GABAergic neurons there seems to be a change also in activity of “ipsi-unidentified” neurons. The size of this effect appears to be of the same order as the increase for ipsi-GABA, but according to the author's statistics - completely not significant. As the authors mention, this result has an important implication to the hypothesis that GABA-ergic activation primarily affects the neural activity on the contra-lateral side. Therefore, the effect of GABAergic activation on the unidentified cells (presumably glutamatergic cells) should be quantified in much greater detail and shown on the level of individual cells as well as population-averaged PSTHs.

In the revised legend we clarify that Revised Fig. 3 shows only *local* activity (i.e., ipsilateral to the side of light delivery) and that “ipsi.” and “contra.” refer to trials in which the mouse chose the ipsilateral or contralateral port (not to the ipsilateral or contralateral SC; p. 12, lines 283–284). To quantify the effect of GABAergic activation during the choice epoch in greater detail, we now show a) distributions of firing rates across trials, separately for ipsilateral and contralateral choices and for light-on and light-off trials, for an example GABAergic and unidentified neuron (Revised Fig. 3b), and b) average PSTHs for the populations of unidentified neurons that increased or decreased in response to light (Revised Fig. 3g,h; quantified at the level of individual neurons in Supplementary Table 2 and at the population level in Fig. 3c), separately for light-on and light-off trials. We thank the reviewer for suggesting this analysis; along with the comparison between population PSTHs for ipsilateral and contralateral choices (see response to Comment 10), these results strengthen our key claim that GABAergic SC neurons do not suppress the activity of local premotor output neurons.

9. Supplementary Table 2 is not very informative and should be replaced by a figure where population-averaged PSTHs of the corresponding groups of neurons are shown, in order to assess the magnitudes and the timescales of the effects.

As described immediately above (response to Comment 8), we have now included population-averaged PSTHs of the groups of unidentified neurons that exhibited increases or decreases in activity in response to light, separately for light-on and light-off trials (Revised Fig. 3g,h). In addition, we have done the same for GABAergic neurons that exhibited increases in response to light (Revised Fig. 3f); we do not include the PSTH for the single GABAergic neuron that exhibited a decrease in response to light. We believe that the information in Supplementary Table 2 complements Revised Fig. 3b, f-h by quantifying the effect of light at the level of individual neurons and therefore believe this table is worth retaining in the revised manuscript.

10. Are those cells that are inhibited (Supplementary Figure 4) by GABAergic photoactivation are special in any way in terms of their PSTH-response profile?

In New Supplementary Fig. 7b, we show the task-relevant population-average PSTHs for those neurons that are inhibited by light during the task (Revised Fig. 3c,h; we suggest that this population is more relevant to this question than the slightly different population shown in Original Supplementary Fig. 4 (now Supplementary Fig. 6), because they were inhibited by light during behavior). This population of neurons appears to exhibit a higher firing rate during choice on ipsilateral than contralateral trials, consistent with the fact that these neurons are presumably inhibited by GABAergic SC neurons, which exhibit a higher firing rate during contralateral than ipsilateral choice (Fig. 4; p. 17, lines 409–416). In addition, it shows that those neurons that are locally inhibited by GABAergic neurons are not likely to be premotor output neurons, which typically exhibit contralateral preference (p. 17, lines 416–418).

11. Are those cells that are excited (non-directly) by GABAergic photoactivation have particular task-related response profiles?

Because we were conservative in which neurons were identified as GABAergic – we minimized false positives at the expense of “missing” some GABAergic neurons (Supp. Fig. 5) – it is possible that some of the unidentified neurons that appear to be non-directly excited by GABAergic photoactivation are actually GABAergic. We therefore urge caution in interpreting results from this group of neurons. Nevertheless, we examined task-relevant population-average PSTHs for those unidentified neurons that are excited by light during the task (New Supplementary Fig. 7a). This population of neurons (13/51 neurons, Supp. Table 2) appears to exhibit a higher firing rate during choice for contralateral than ipsilateral choices, even more so than does the population of unidentified neurons with activity unaffected by light (New Supplementary Fig. 7c; 27/51 neurons, Supp. Table 2; p. 17, lines 411–416). This finding makes sense, since these neurons are presumably either GABAergic themselves (but were conservatively not classified as such) or are indirectly excited by activation of GABAergic SC neurons, which exhibit a higher firing rate during contralateral than ipsilateral choice (Fig. 4c, bottom).

12. I don't suggest new experiments, but in case the authors already have this data - it would be nice to see if indeed ipsi-lateral activation of GABAergic neurons inhibits the activity of neurons on the contra-lateral SC.

We agree that this is an interesting question, but unfortunately we did not perform these experiments. As noted above (response to Comment 7), we plan to thoroughly examine the functional role of intercollicular neurons in a future study. We suggest that it would be particularly valuable to perform the stimulation/behavior/recording studies recommended by the reviewer during spatial choice (p. 23–24, lines 576–579).

13. The authors suggest that ipsi-lateral GABAergic inactivation has no effect on the activity of ipsi-lateral principal cells, but instead might affect the contra-lateral SC. This might be inconsistent with long-range inhibition hypothesis: According to this hypothesis, one may expect that if the activity of the contra-lateral neurons is biased by ipsi-lateral GABAergic photoactivation, they should in turn inhibit the SC neurons (via inhibitory neurons projecting from the contra to the ipsi side). The authors might want to explore this point using their network-model.

We assume that the reviewer means “ipsi-lateral GABAergic activation,” since we do not present the results of GABAergic inactivation experiments here. In our model, activating ipsilateral inhibitory neurons transiently inhibits ipsilateral excitatory neurons and, via intercollicular projections, inhibits excitatory and inhibitory neurons in the contralateral SC. The inhibitory intercollicular projections from the contralateral SC are therefore inhibited, resulting in net excitation of the ipsilateral SC. Although this prediction is beyond the scope of our narrowed claims in the revised manuscript, the fact that our model shows that ipsilateral SC is more excited relative to the contralateral SC when *I* cells are activated is consistent with our key observation that unilaterally exciting GABAergic SC neurons promotes a contralateral choice bias (Revised Fig. 2a,b).

Minor comments:

14. The authors might compare and contrast bilateral models of ALM (Li/Daie et al 2016; Inagaki et al 2020) and keep in mind that the SCm is bidirectionally connected to frontal cortex.

In the revised Discussion, we briefly compare bilateral models of SC and ALM:

Interestingly, similar to the SC, unilateral inhibition of ALM principal neurons during action selection disrupts contralateral choices⁶³. However, if inhibition of ALM is limited to early in the selection process, circuit dynamics can be recovered by activity from the opposite ALM and accurate action selection is preserved⁸⁹, while similar early inhibition of SC disrupts choice²⁹. Thus, the two ALM hemispheres appear to cooperate, while the two sides of the SC appear to compete, to mediate left-vs-right spatial choice. (p. 24, lines 591–596)

The relationship between SC and ALM models deserves much more extensive treatment, but given that we have significantly reduced discussion of decision-making models in the revised manuscript (see response to Comment 36), we believe this small addition fits within the Discussion while adding value.

15. Introduction: Most research about the role of SC in spatial cognition was conducted in primates. It would help the reader if the authors specify which papers that are mentioned in the introduction are based on SC studies in rodents.

In the revised Introduction, we have segregated references based on which species they refer to (e.g., p. 3, lines 53–60).

16. Page 4 (line 85): The functional outcomes of the two hypotheses are not very clear, especially for the second hypothesis. They become more clear when the authors refer to Figure 1, but these happens only at the very end of the introduction.

We have significantly revised the Introduction to focus only on the hypothesis that GABAergic SC neurons suppress local premotor output neurons during spatial choice (see response to Comment 29), and we now refer to the hypotheses and predicted results in Fig. 1 as early as possible (p. 5, lines 102–104).

17. *Figure 1b,d - what is represented by triangles, circles? The logic why some of the lines are in bold is not immediately clear.*

In Revised Fig. 1b we have placed a box around the key indicating that circles and triangles represent populations of GABAergic and premotor output neurons, respectively. In addition, we have indicated in the legend that line thickness corresponds to the hypothesized level of activity (p. 7, lines 165–166).

18. *It is unclear how the hypotheses about local versus long-range inhibition map onto winner-take-all/race-to-threshold models mentioned in the introduction and the discussion. The authors mention in the discussion that race-to-threshold models “do not require such inhibitory interactions”, but it would help the reader to know that earlier - in the introduction, when the two alternatives are described.*

We have significantly overhauled the revised manuscript to focus on the experimental results rather than on how they inform theoretical models of decision making and have therefore removed our discussion of these models (see response to Comment 36). In general we have attempted to introduce discussion points earlier. For example, in the Introduction we now describe how the results of Wang et al. (2020) support our premise (p. 5, lines 98–104), before comparing our results with theirs in the Discussion (p. 24–25, lines 599–619; see response to Comment 39).

19. *Does the effect of photostimulation of GABAergic neurons on reaction time consistent more with race-to-threshold or winner-take-all models?*

We have clarified in the revised Results that the effect of photostimulation of GABAergic neurons is consistent with competitive interactions between the left and right SC (p. 8, lines 185–191). However, we do not claim that these competitive interactions are directly mediated by GABAergic SC neurons, and, as noted by Reviewer 4 (Comment 36), even race-to-threshold models may be mediated by competitive interactions. Thus, we do not attempt to use our results to differentiate between the models.

20. *Fig. 2c. The effect of ipsilateral muscimol injection (ipsi-lateral bias) can be in principle attributed to an increase in the effect of long-range inhibitory projections from the contra-lateral side. The authors should discuss this possibility.*

We believe the reviewer is suggesting that, if GABAergic SC neurons act by inhibiting the contralateral SC during spatial choice, muscimol could be activating those GABAergic receptors that are endogenously activated by intercollicular projections from the contralateral side. Therefore, our muscimol results (Revised Fig. 2c,d) are consistent with an intercollicular role for GABAergic SC neurons. We agree with this interpretation, although given our refocusing of the manuscript on our experimental results and the removal of our terminal stimulation data, we have significantly reduced discussion of intercollicular inhibition in the revised manuscript (see response to Comment 37).

21. *The result in Fig.2e is almost trivial, and does not advance the paper, which is focused on GABAergic neurons. I suggest moving it to the supplementary, and discussing more the difference between Fig.2a and Fig.2e.*

We have moved Original Fig. 2e,f to Supplementary Fig. 3, and discuss how these results relate to those of Revised Fig. 2a,b (p. 10, lines 235–237), to keep the focus of the paper on GABAergic SC neurons.

22. *Page 11 (line 254): “GABAergic firing rates” should probably be “GABAergic neurons firing rates”.*

We have edited this sentence in the revised manuscript (p. 11, lines 252–253).

23. What do the authors refer to as “ipsi” “contra” choices in Fig.3b-c? Does it refer to the selectivity of the neurons?

“Ipsi.” and “Contra” refer to the choice on that trial, not to the preference of the neurons. We have added text to the Results (p. 11, lines 257–260) and revised legend of Revised Fig. 3c (p. 12, lines 283–284) to clarify this point.

24. The authors should show more example neurons in Fig. 4 and also the population average PSTH for GABAergic and non GABAergic neurons.

In Revised Fig. 4e, we now show the activity on contralateral *and* ipsilateral trials for all contralateral-preferring GABAergic neurons. We believe this presentation is even more informative than showing a population-averaged PSTH since it presents the activity of each neuron. Since our unidentified neurons could be either GABAergic or glutamatergic (p. 11, lines 253–257), and our focus here is on the GABAergic neurons, we believe it would detract from our focus to show PSTHs or a heatmap for unidentified neurons in Fig. 4. However, we do show population average PSTHs for specific groups of GABAergic and unidentified neurons in New Supplementary Fig. 7.

25. What is the interpretation of the result presented in Fig.4d?

We have clarified that the results presented in Fig. 4d demonstrate that contralateral choices were more likely on trials with higher firing rates of GABAergic SC neurons (p. 15, lines 372–375), complementing the results presented in Fig. 4c.

26. The authors should discuss if the fact that the GABAergic activity is higher on contra-lateral trials (Figure 4) may be because local GABAergic neurons may reflect the population average activity of local principal neurons (which increase their activity on contra-lateral trials). This can be also explored using the attractor model that the authors present in this paper.

We agree with this point and have clarified in the revised Discussion that “GABAergic SC neurons may integrate the activity of contralateral-preferring SC neurons” (p. 23, lines 574–576; Revised Fig. 7). We have also clarified that, in the model, local excitation from excitatory neurons is required to maintain the activity of inhibitory neurons (p. 19, lines 473–475).

27. Discussion (page 22, 533) - what do the authors refer to as “multiple SC loci”? I found the entire discussion (page 22-23) regarding “winner-take-all” versus “winner-take-most” mechanisms and “SC loci” totally unclear. Specifically, it was unclear what is the difference between these two mechanisms and how does the current data distinguish between them.

We have significantly overhauled the manuscript to focus on the experimental results, rather than on how they inform theoretical models of decision making (see response to Comment 36). In the revised manuscript, we have therefore removed the text describing how our data relate to these models, and we no longer use the term “multiple SC loci” (which we had intended to refer to populations of SC neurons at different locations in the topographic map that represent distinct spatial targets).

28. Discussion (Page 23): the fact that most GABAergic neurons were indifferent to choice, does not necessarily mean that they are not part of the choice-selection mechanism (see Najafi et al Neuron, 2020 “Excitatory and Inhibitory Subnetworks Are Equally Selective during Decision-Making and Emerge Simultaneously during Learning”).

We agree with this point, and have edited this section to read that “while these neurons may nevertheless contribute to choice, they may also be involved in acquiring targets or other SC-mediated behaviors not examined in our paradigm” (p. 26, lines 639–642).

Reviewer #4

In this study, Essig et al aim to investigate the role of GABAergic neurons in SCid in spatial decision making. This question is an important one, and the authors have expended significant effort in attempting to address this question: they use a two alternative forced choice behavior combined with cell-type specific optogenetic manipulation, as well as opto-tagged recordings; kudos on the effort. The authors activate GABAergic neurons in the intermediate and deep SC (SCid) layers and seem to observe a, counter-intuitive, contraversive shift in choice bias. (Past literature has identified GABAergic neurons in the SCid to have both intracollicular as well as cross-collicular projections.) To support their observation the authors make the claim that photoactivating GABAergic neurons mainly affects crosscollicular projections. Further, the authors use optotagged recordings from GABAergic neurons to show that a subset of the tagged GABAergic neurons show activity that correlates with contraversive choice. Finally, the authors attempt to support their claim of GABAergic inhibition working via the crosscollicular projections by constructing an attractor model.

However, the results and the experimental techniques used in the study do not support the claims and the interpretation of the results that the authors lay out in the manuscript. This discrepancy between results and interpretation significantly impacts and narrows the scope of the results. A combination of potentially improperly analyzed data, ambiguous (and not clearly interpretable) results, sub-par writing, and overextended claims in terms of the circuitry as well as modeling make the paper unsuitable for publication in its current form. (Some of these concerns appear to have already been raised by previous reviewers, and the authors do not seem to have addressed them satisfactorily.) Improved writing and scholarship, better analysis, significantly reduced claims re theoretical/modeling issues, coherent flow, and a tight presentation of a story focused primarily on the experimental questions and results – all well within the capability of the authors - will address many of these concerns. However, even with that done, the fundamental scientific issue at the very heart of the study will still remain unresolved/uninterpretable from the data presented, which is what significantly dampens enthusiasm for the paper, unfortunately. Details below.

Detailed comments/The story and findings.

The clearest story that appears to be extractable from the results is as follows. (This required multiple reads of the difficult-to-follow manuscript.) Concerns are highlighted in blue.

We thank the reviewer for their comments and specific recommendations on how to improve the manuscript. As described below, we have significantly overhauled the manuscript according to these suggestions, which we believe has made the story – which the reviewer correctly understood – much easier for readers to follow.

29. The role of SCid GABAergic neurons in spatial decision-making is not clear (although see very recent study by Wang...Krauzlis 2020, J Neurosci). Based on the wealth of literature about the SCid including results from manipulating excitatory neurons in SCid with electrical stimulation and muscimol delivery, results from slice experiments as well as our knowledge of SCid anatomy, a traditional expectation would be that activating GABAergic neurons in one hemisphere would suppress nearby excitatory neurons, resulting in suppression of contralateral behavioral choices/responses.

The reviewer's suggestion was indeed our intended premise. We believe that this premise has been further clarified by our overhaul of the manuscript. In particular, Revised Fig. 1 now emphasizes the single hypothesis that GABAergic SC neurons locally suppress premotor output, rather than presenting multiple competing hypotheses to be tested. Indeed, this hypothesis guided the initial design and execution of our experiments and we agree that focusing on it clarifies the rationale for our approach.

30. In order to test the role of SCid GBAergic neurons, the authors stimulate GABAergic neurons in GAD-cre mice. [This is already problematic for two reasons: 1) Activation experiments only reveal sufficiency, not the "role" of the neurons in normal behavioral function.

We agree with this comment. In the revised Abstract we state that we examine the role of GABAergic SC neurons in spatial choice not only by perturbing their activity in the context of behavior, but by our entire array of approaches: perturbing and recording activity during behavior, and using our attractor model (p. 2, lines 26–30). We further

clarify in the revised Results, when introducing our photoactivation experiments, that these experiments “begin to test the function of inhibitory SC neurons during choice” (p. 6, lines 135–138).

31. Specifically, activation of GABAergic neurons within circuits in the cortex as well as in SCid is an experimental “trick” used to shut down nearby excitatory neurons as opposed to investigate the “role” of the GABAergic neurons themselves. Not only do the authors not acknowledge this, they have on more than one occasion, misinterpreted the Wang...Krauzlis 2020 paper by referring to the results as showing the function of GABAergic neurons, whereas the original authors of the paper state clearly that the results are better viewed as the consequence of silencing excitatory neurons. Nonetheless, activating GABAergic neurons and examining the effect of this manipulation on neural responses and behavior is a fine thing to do. So let us proceed.]

We recognize that activating GABAergic neurons is a common approach for achieving local inhibition and we have clarified in the revised Introduction that this was part of the rationale for our hypothesis (Revised Fig. 1b,c; p. 5, lines 98–109). We also describe this approach in the revised Results and Discussion as well (p. 6–7, lines 144–149; p. 24–25, lines 601–605). In addition, we have revised our description and discussion of Wang et al. (2020)’s results, as detailed in our response to Comment 39.

32. Contrary to expectation, they potentially find that activation of GABAergic neurons promotes contralateral responses. [I say ‘potentially’, because here, as in most figures, the unit of statistical analysis, i.e., n, that they use is a session. This is not appropriate, and is also not the convention. The appropriate n should be number of animals (and this should be specified clearly within each figure legend, as opposed to once in the Methods). The reason this issue is critical is: say, the number of animals used in Figure 2a (the lynchpin result of the paper) was 8, with an uneven number of sessions per animal. It is possible that the result was robust in one or two animals and was not apparent in the others. If, out of the total number of sessions reported, a disproportionately large fraction of them was from these two animals, this could likely result in their finding being not true, statistically. Unless the data are analyzed and plotted in terms of n=8 animals (rather than the current n=96 sessions), it is impossible to assess the validity of the finding. This analysis is easy to re-do, and for the purposes of this review, let us proceed with the assumption that the finding will still stand true once the analysis and plotting are re-done correctly – i.e., GABAergic neuronal activation promotes contralateral choices.]

In the revised manuscript we have replotted the data to show the effects by mouse and we report the statistics and “n”s with respect to mice (Revised Fig. 2a,b; Revised Supp. Fig. 3; p. 8, lines 182–185; p. 9, lines 205–213; p. 10, lines 228–232). The reviewer is correct that the findings stand true when plotted and analyzed by mouse. However, after carefully considering this thought-provoking recommendation, it is not obvious to us that the unit of analysis should be mice rather than sessions, given the variability in experimental conditions, and our results, across sessions from the same mice (Makin and De Xivry, 2019). In particular, in the Gad2-Cre mice (Revised Fig. 2a,b) we photoactivated at different depths across sessions, presumably activating different populations of neurons. We therefore conservatively also plot and analyze data with respect to sessions (Revised Supp. Fig. 2). In addition, Supplementary Table 1 shows that the number of sessions performed was relatively balanced across these mice, and no Gad2-Cre mice exhibited an ipsilateral bias (Revised Fig. 2b).

33. As a sanity check in the face of this unexpected/puzzling result, the authors do two experiments: silencing excitatory neurons pharmacologically, and activating excitatory neurons optogenetically (in VGLUT-cre mice), and they find the expected result in both cases. [This was not really necessary to do, in my opinion, and could easily be moved to supplementary data, because the muscimol experiments and the scientific outcome of the optogenetic activation of excitatory neurons are very well established.]

To better focus the manuscript on the role of GABAergic SC neurons, we have moved the results of the Vglut2-Cre mice to Revised Supplementary Fig. 4. However, even if the result is not surprising, we believe it is important to highlight that muscimol, which provides direct local inhibition to premotor output neurons (in addition to other nearby neurons), results in the opposite choice bias as photoactivation of GABAergic neurons and therefore retain it in Revised Fig. 2c,d. We have clarified the purpose of the muscimol experiments was to “examine the behavioral effects of local... inhibition in mice performing the task” (p. 9–10, lines 223–226).

34. To investigate this central puzzling result (Fig. 2A), the authors perform opto-tagged recordings. They verify this approach in Figure 3, show in Figure 4 that a greater fraction of opto-tagged GABAergic neurons show responses correlated with contralateral choice than ipsilateral choice, and show in Figure 5 that greater activity of opto-tagged GABAergic neurons correlates with higher likelihood of contralateral choice. [I appreciated their demonstration in Fig. S3 that in the SCid, attempting to classify GABAergic neurons simply on the basis of waveform shape is not effective]. [For clarity, these figures could be grouped into one or at most two figures, with the rest of the info moved to supplementary. Importantly, the motivation for and description of these results can be described more succinctly and clearly]

We have considered regrouping Figs. 3-5. However, given our additions to these figures based on the totality of reviewer comments, and given that they each show the results of different experiments and analyses, we believe that it is most appropriate to retain them as separate figures in the main manuscript. Our significant rewriting of the Results section more succinctly and clearly describe the motivation for and description of these results (e.g., p. 10, lines 243–245; p. 15, lines 372–375).

35. The authors then hypothesize that one account for this unexpected finding could be greater strength of intercollicular inhibition compared to intra-collicular inhibition (Inter-collicular inhibition is known, anatomically, to exist, as they point out in the intro). They illustrate this idea using a computational model (Figure 5). [This is the only utility I see of the model as it has been presented – as a quantitative illustration of this idea of greater intercollicular versus intracollicular inhibition efficacy. This is essentially a sophisticated curve-fitting exercise, in the sense that a model has been built to fit/explain their data. However, this is only one half of the utility of any good modeling effort because one can always fit a “curve” (model) to data. The true worth of a model is measured by the testable predictions it makes, and the testing of the predictions. Since the authors do not do this, the model serves only to account for their data, meaning that any number of alternative accounts may also be valid.

In the revised manuscript we clarify that the purpose of the attractor model is “to determine if mechanisms within the two SCs (i.e., without invoking extracollicular structures) are sufficient to reproduce a contralateral choice bias when local and long-range inhibitory neurons are photoactivated” (p. 19, lines 462–465) and that we first fit the model to reproduce the contralateral choice bias observed in our GABAergic activation (i.e., stimulation/behavior) experiments (p. 21, lines 506–508). As the reviewer notes, the primary value of the model in this manuscript – particularly since we have removed the terminal stimulation experiments from the revised manuscript (see response to Comment 37), which were intended to test the model’s predictions – is that it quantitatively demonstrates the intuitive idea that our results can be explained by GABAergic SC neurons providing functionally stronger intercollicular inhibition than local inhibition (even though both forms of inhibition are present). Finally, we agree that the model does not rule out alternatives, and we clarify that our model “provides one explanation” for our results (p. 21, lines 525–527) and “does not rule out a role in spatial choice for long-range projections of GABAergic SC neurons to other brain regions, since it was explicitly designed to examine how our results could be explained with only SC circuitry” (p. 24, lines 584–587). We believe these changes clarify the intended role of the model in this manuscript.

36. Further, their setting up of two ‘competing’ modeling perspectives is largely a straw-man argument. Although the race models (or accumulation to bound models) proposed originally did not include any inhibition (and involved independent accumulators), modern views of these models includes inhibition, and has done so for at least 10 years now. There are many review papers, particularly on perceptual decision-making that cover this ground (for instance, Churchland et al, 2012, Curr Opin Neurobiol; Mysore et al 2020, eLife). In this context, the fact that the authors heavily infuse ideas about models of decision-making throughout their paper, comes across as not appropriate. In fact, the entire set-up of the paper, right at the outset of the introduction, is that the study is going to help us understand models of decision-making better, which is disingenuous at best- I strongly recommend that the authors simply remove lines 37-49, and begin the intro at “Spatial choice-...”, which would allow the paper to be focused on the experimental question and results. (Potentially, the narrative laid out in the points here could be one that the authors may consider using.) They would also need to significantly streamline the Intro to keep this focus on the experimental question. Any mention of modeling should be done in a limited manner in the Discussion.]

Guided by the reviewer's suggested narrative we have significantly overhauled the manuscript to focus on the experimental results, rather than on how they inform theoretical models of decision making. In particular, we have deleted most of the first paragraph of the original Introduction, allowing us to focus more clearly on the neural basis of spatial choice. The Introduction now begins:

Decision making is fundamental for adaptive behavior and examining its neural bases offers insight into circuit mechanisms for cognitive processes^{1,2}. An effective strategy to study decision making circuits is to examine the activity of specific types of neurons in conserved brain areas during ethologically-relevant behaviors⁵. Spatial choice—selecting where to attend or to move—is ideal for this purpose: it is an adaptive form of decision making that is critical for survival and, across primates⁶⁻⁹, cats¹⁰ and rodents¹¹, depends on computations performed by the intermediate and deep layers of the midbrain superior colliculus (SC).

We believe the first two sentences (preceding “Spatial choice—...”) are necessary to provide the general reader with the rationale for examining spatial choice.

In addition to this change in the Introduction, we have significantly reduced and edited our discussion of decision-making models throughout the revised manuscript, including in the last sentence of the Abstract, first sentence of the Discussion, and several other places (p. 2, lines 36–38; p. 6, lines 125–126; p. 22, lines 530–532).

37. The next obvious step would be to test experimentally, this hypothesis. The way to do this would be to activate cross-hemispherically projecting GABAergic fibers to test if this can promote contralateral choices. This is the critical experiment to test their hypothesis. [I originally thought that they had not done this experiment and was puzzled by why they hadn't. It turns out that they actually did do this experiment, but the results are only found in Supplementary Figure 7 and seem to only be brought up in the Discussion! It turns out that the results do not appear to support their hypothesis. Out of 11 sessions, only 3 show the expected result. This is problematic for a few reasons: 1) As in previous figures, this should be analyzed and plotted in terms of number of animals and not number of sessions. How many animals were tested here? What is the result if data are plotted with n=animals? 2) If their hypothesis is true, most sessions should show an effect in the predicted direction, which does not seem to be the case. Therefore, this critical experiment seems to not support their hypothesis for why activating GABAergic neurons would promote contralateral choices, leaving us to consider alternative explanations (more on this in point #8). As a side note, this very issue was brought up also by a previous reviewer. The authors' response seems to be that this is 'preliminary' data and to move it to supplementary, neither of which action actually responds to the question! The notion of 'preliminary' data in a paper is not meaningful. And moving it to supplementary and seemingly brushing it out of view is also not appropriate.]

We have removed the data on terminal photoactivation to focus the revised manuscript on our main experimental findings. We considered these data to provide preliminary support for the predictions of the model – in particular, photoactivating terminals had a very clear effect on reaction times, indicating their involvement in this spatial choice task – but, as noted by Reviewer 3, we likely provided insufficient light power to fully photoactivate GABAergic SC terminals (see Comment 7). Thus, instead of presenting these data here, we will build on our current experimental and modeling results to focus on the functional role of intercollicular neurons (GABAergic and glutamatergic) in a future study (p. 23–24, lines 576–579).

38. At this point, we are left with the following scientific results: That activation of inhibitory SCid neurons can cause promotion of contralateral choices, that activity of many GABAergic neurons is correlated with contralateral choice, that this could be because of stronger inter-collicular inhibition compared to intracollicular inhibition, but that upon testing directly (with fiber activation), this does not seem to be the explanation. Because the hypothesis does not seem to be supported, the core findings then are simply the first two, which boil down to “activation of GABAergic neurons can promote contralateral choices, contrary to traditional expectation”. As I said at the outset, even if all the issues with analysis and writing are addressed, this remains seemingly the primary scientific advance of the study, which is incremental at best. Given that we know that cross-hemispherical GABAergic projections exist, that they can promote contralateral choice is not surprising. And I would like to point out again that this has really nothing to do with two different ‘opposing’ models, or about achieving deep insights into circuits of models for decision-making. This is simply an unexpected experimental finding.

We clarify in the revised Abstract that “our results demonstrate that GABAergic SC neurons do not locally suppress premotor output, suggesting that functional long-range inhibition instead plays a dominant role in spatial choice” (p.

2, lines 30–34). This result was indeed unexpected, as it was inconsistent with the hypothesis that GABAergic SC neurons function to locally suppress premotor output (Revised Fig. 1b). In the revised Introduction, we clarify that this hypothesis was well-premised on the results of previous studies using photoactivation of GABAergic SC neurons (Ahmadlou *et al.*, 2018; Wang *et al.*, 2020) as well as on the observation of ipsilateral-preferring SC neurons (Horwitz and Newsome, 2001; Hirokawa *et al.*, 2011; Felsen and Mainen, 2012) (p. 5, lines 100–107). In addition, although the anatomy of GABAergic SC neurons has been described (Mize, 1992; Behan *et al.*, 2002; May, 2006; Sooksawate *et al.*, 2011), we clarify that, to our knowledge, no studies have examined their functional role in decision making by directly recording and manipulating GABAergic SC activity during behavior (p. 3, lines 70–72; p. 6, lines 132–135) (and very few have examined the contribution of inhibitory neurons in any brain region to decision making (Najafi *et al.*, 2020)). Our findings therefore inform our understanding of how inhibitory neurons function within neural circuits mediating spatial choice (p. 22, lines 544–546) and will serve as the basis for future studies (p. 23–24, lines 576–579).

We agree that we have not demonstrated a functional role for intercollicular inhibition in spatial choice and have therefore removed these data from the revised manuscript. However, we do not believe that our results rule out such a functional role, only that they are not sufficiently conclusive. We therefore do not believe we can justify stating in the revised manuscript that intercollicular inhibition cannot explain our results; instead, we will examine this possibility in future studies (p. 23–24, lines 576–579; see response to Comment 37).

39. In this context, then, an important point to discuss would be why they find this unexpected result, whereas others (specifically Wang ... Krauzlis 2020) have found the expected result? This discussion becomes extremely important if the per animal re-analysis of Figure 2 pans out. For instance, could the difference in results simply be the result of chance differences in the placement of the virus and the proportion of ipsi versus contra projecting GABAergic neurons that are impacted? Could there be differences in the subtypes of GABAergic neurons recruited by using GAD-cre mice vs. VGAT-cre mice (as in Wang et al), coupled with differences in the projection patterns (local versus cross-hemispherical) of subtypes? These possibilities could account for all of their results (Fig.2, Fig 5, Supp Fig 7), especially considering that the results in these figures do not overwhelmingly support their claim, and that there are many instances of the expected result as well. As the data in this paper do not strongly support one (their) explanation over others, it is difficult to conclude definitively. As things stand, then, the best case that can be made from these data, it seems, is that it is possible for GABAergic neurons to promote contralateral choices. This is, unfortunately, not a significant advance nor new or unexpected.

In the revised Discussion we extensively compare our results to those obtained by Wang et al. (2020):

A recent study found that unilateral photoactivation of GABAergic SC neurons affected performance on a visual detection task for stimuli presented to the contralateral, but not ipsilateral, visual hemifield³⁰. These results are consistent with local suppression of neuronal activity representing contralateral space – the authors’ intended, and our hypothesized, effect of photoactivating GABAergic SC neurons – raising the question of why we obtained a seemingly opposite effect on spatial choice (a contralateral choice bias) using essentially the same photoactivation method. Although our study was superficially similar to that of Wang et al. (2020) – e.g., both examined the role of the SC in mice performing a decision-making task – differences between how the tasks engage SC circuits are likely critical. First, the task used by Wang et al. (2020) was a visual change detection task, and the largest effect of photoactivation was observed when visually-evoked activity in the SC was highest. It is possible that the same population of SC neurons that we observed to be locally inhibited by GABAergic SC activation (Fig. 3h, Supplementary Table 2), but which were not responsible for contralateral choice (Supplementary Fig. 7b), are required for visual change detection. GABAergic SC neurons project to the superficial SC and may suppress visual responses^{42,86,90}, providing another potential pathway for disrupting contralateral visual representations. Second, the task used in Wang et al. (2020) did not require a spatial choice; mice reported a change in either visual hemifield with a single motor output. If the authors had instead required the selection of a directional movement based on a visual stimulus, or if our study had required a non-directional response to a lateralized stimulus, we might expect the results of the two studies to have been similar. (p. 24–25, lines 599–619)

We also note that Wang et al. assumed that photoactivating GABAergic SC neurons would suppress local activity, as GABAergic photoactivation has been shown to do in other studies. However, unlike in our study (Fig. 3; Supp. Fig. 6), this assumption was not tested by recording local activity while photoactivating GABAergic SC neurons during the task. While other methodological differences (e.g., different mouse lines, head-fixed vs. freely-moving behavior)

may further preclude direct comparisons between our results and those of Wang et al., it is most likely that the difference in experimental paradigms described in the revised Discussion (copied above) play the largest role in our seemingly opposing results.

Ultimately, the totality of our results—including perturbing and recording activity during spatial choice, and supported by our modeling results—demonstrate that the activity of GABAergic SC neurons during spatial choice promotes the selection of contralateral targets, and that these results are unexpected. By showing that GABAergic SC neurons do not locally suppress premotor output, our study prompts a rethinking of their traditional role and serves as a basis for future studies to elucidate SC circuitry underlying spatial choice.

40. As asides, the title is also misleading; their use of “long-range” inhibition should be replaced by intracollicular inhibition throughout because the former phrase is misleading/ambiguous. The title could at best read “Inter-collicular inhibition can mediate goal-directed decision making”.

We have re-titled the manuscript to “Inhibitory neurons in the superior colliculus mediate goal-directed decision making,” which we believe provides a clear, concise and accurate description of our study. While our results demonstrate that GABAergic SC neurons do not suppress local premotor output during spatial choice, suggesting a role for at least one form of long-range inhibition in mediating spatial choice (intracollicular, intercollicular and extracollicular; Revised Fig. 7), we agree that the manuscript does not demonstrate distinct contributions of any of these forms (as described in detail in our responses to Comments 3-5, 35, 37).

41. Others: There are many other issues with writing, analysis, and scholarship. Only a few are listed here and below because the scientific and structural concerns are more imposing: a) Figure legends are not clear. They should be clear descriptions of the panels as opposed to discussion of the meaning; legends should include stats.

We have revised the figure legends to clarify the descriptions of the panels without discussing their meaning, and to include the relevant statistics (e.g., p. 12, lines 283–295). We have also attempted to address any other potential issues with writing, analysis and scholarship.

42. The need for a GLM type quantification of results in Figure 2 (with the poorly explained betas) was unclear. Similar work in monkeys simply fit sigmoids and quantify the half-max or threshold to assess leftward rightward shifts.

We have applied both sigmoidal fits and logistic regression analyses to the data in this study and in previous studies using the same behavioral task (e.g., (Stubblefield *et al.*, 2013; Thompson *et al.*, 2016)). We have found that logistic regression more accurately reflects shifts in the psychometric function, in part because it does not assume that the slopes of the psychometric functions under comparison are unchanged. Nevertheless, the values obtained with each of these methods are highly correlated (Reviewer Fig. 1) and the method of analysis would not change our overall results. Since β s obtained with regression are even more conservative than values obtained from sigmoidal fits for larger shifts (Reviewer Fig. 1), and since logistic regression is a standard method and has been used to quantify how perturbing neural activity influences decision making (e.g., (Salzman *et al.*, 1992)), we use β s obtained with logistic regression here (p. 36–37, lines 892–900). In the revised Methods we have also clarified our description of this analysis (p. 35–36, lines 872–884) and stated that it yields similar results to sigmoidal fits (p. 36–37, lines 895–897).

Reviewer Figure 1: Comparison between bias calculations from sigmoidal shifts in task performance and $\beta_{manipulation}$ from logistic regression for each session (gray dots) and each mouse (black diamonds) from GABAergic photoactivation experiments (Revised Fig. 2a,b; Supp. Fig. 2a).

43. *The discussion does not include treatment of the central issue, described above in point #8.*

As detailed in our response to Comment 39, we have included a new section of the Discussion describing our results in the context of those of Wang et al. (2020).

44. *Supp Fig 3 is very jumbled; panels need to be reorganized.*

Our understanding of the reviewer's concern is that Original Supplementary Fig. 3 is organized in multiple columns (a/b/c, d/e, and f/g). We have now reorganized the panels in this figure (now Revised Supplementary Fig. 5) such that each row follows from the one above it, which we hope addresses this issue.

Additional comments:

45. *Figure 1a: Assumption that photoactivation of GABAergic neurons mainly has an effect of inactivating the inter-SC projecting neurons is unsubstantiated.*

We have simplified the revised manuscript to only present the hypothesis that GABAergic SC neurons locally suppress premotor output neurons and have therefore removed the original panel showing photoactivation of intercollicular projections. More generally, the inset in Original Fig. 1c was not intended to reflect an assumption, but rather our hypothesis that photoactivating GABAergic neurons will locally suppress premotor output neurons, despite the existence of GABAergic intercollicular projections, leading to the predicted ipsilateral choice bias. We have changed the inset by removing the anatomical projections to make it consistent with the inset in Revised Fig. 2a.

46. Lines 67-70: *The authors claim that traditional methods are unable to isolate GABAergic neurons to dissociate contributions of intrinsic and extrinsic SC inhibitory mechanisms. - However, the techniques that the authors use is basically to infect “all” SC GABAergic neurons, and also does not distinguish between inter-collicular and intra-collicular inhibition. (See Wang Krauzlis, 2020)*

We have edited this text to clarify our intended point: that the advantage of our approach is to distinguish between intrinsic inhibition provided by GABAergic SC neurons and extrinsic inhibition provided by other brain regions (e.g., substantia nigra pars reticulata; p. 4, lines 76–83).

47. Lines 75-76 - *It seems that the authors have misinterpreted a recent study (Wang Krauzlis, 2020) which uses similar methods of activating GABAergic neurons in the SC. Since the techniques are the same, why do the authors interpret their photoactivation as inter-collicular whereas the Wang Krauzlis, 2020 is interpreted as intra-collicular?*

We have deleted the reference to local inhibition here (p. 4, lines 86–89). As described in our response to Comment 39, we have extensively examined the differences between our study and Wang et al (2020) in the revised Discussion.

48. Lines 102-103: *The authors provide no experimental (anatomical) evidence that local GABAergic neurons exert a greater inter-collicular inhibitory effect rather than intra-collicular inhibitory effect.*

We have clarified in the revised Introduction that anatomical evidence exists that GABAergic SC neurons project intra-collicularly, inter-collicularly and extra-collicularly (p. 4, lines 84–92). However, anatomical evidence alone does not address the functional role of GABAergic SC neurons in spatial choice. To do so, we must examine their activity *in vivo* while targets are selected (p. 4, lines 92–97).

49. *One way could be to reconstruct the terminal fields of the inhibitory neurons.*

We agree that reconstructing terminal fields would be illuminating and add to the literature on this topic (Sooksawate *et al.*, 2011). However, doing so would require significant additional experiments more suitable for a manuscript focusing on SC anatomy, rather than the current manuscript focusing on the functional role of GABAergic SC neurons in spatial choice.

50. *The other way could be to inhibit activity of inter-collicular GABAergic projections during behavior. This would give a direct answer to the role of inter-collicular GABAergic projections in selection.*

Indeed, this is an excellent idea and we attempted experiments to inhibit intercollicular terminals during behavior (via Arch-mediated photoinhibition). However, we did not achieve sufficient terminal inhibition, possibly due to insufficient light power (see Comment 7) and possibly due to documented issues with terminal photoinhibition (Mahn *et al.*, 2016; Lafferty and Britt, 2020). As described in our response to Comment 37, in the revised manuscript we have removed the results of intercollicular GABAergic terminal activation experiments, and we plan to focus future studies on the functional role of intercollicular neurons in spatial choice (p. 23–24, lines 576–579), including by inhibiting their terminals.

51. *Supplementary Figure 7: The results of this data are inconsistent and do not support the results of Figure 2a. It seems that the weak +ve correlation is driven mainly by 1-2 behavioral sessions. Please also see previous comments on the need for re-analyzing and re-plotting several figures with comparisons one across mice as opposed to across sessions.*

As described in our response to Comment 37, we have removed the results of intercollicular GABAergic terminal activation experiments and Original Supplementary Fig. 7; we will focus on the functional role of intercollicular neurons in spatial choice in future studies (p. 23–24, lines 576–579).

52. *Figure 1b: Assumption that using muscimol has mainly a local GABAergic effect is also unsubstantiated (lines 180-181). a. Muscimol works by activating post-synaptic GABAergic receptors and not via activating GABAergic neurons. Thus, delivering muscimol basically reduces activity of all cells with active GABAergic receptors irrespective of whether these receptors form synapses with local intra-collicular interneurons or with inter-collicular projections or inhibitory projections from outside the SCid.*

We have clarified in the revised manuscript that our purpose in using muscimol was to “examine the behavioral effects of local... inhibition in mice performing the task” (p. 9–10, lines 223–226). We interpret the fact that muscimol resulted in an ipsilateral bias, while photoactivation of GABAergic SC neurons did not, as evidence that the inhibition provided by GABAergic SC neurons is predominantly non-local, rather than serving to locally suppress premotor output, during spatial choice (p. 10, lines 237–240).

53. *Activating GABAergic post-synaptic receptors or activating pre-synaptic GABAergic neurons (using light) to turn-off upstream excitatory neurons is a common technique used across the field. In these studies, very rarely are comments regarding the functional role of GABAergic neurons made.*

As described in our response to Comment 31, we have clarified in the revised manuscript that activating GABAergic neurons is a common approach for achieving local inhibition (p. 5, lines 98–104; p. 6–7, lines 144–149; p. 24–25, lines 601–605). In addition, as described in our response to Comment 30, we agree that activating GABAergic neurons is insufficient for identifying their functional role. Here, we identify the functional role of GABAergic SC neurons by perturbing and recording activity during behavior, in conjunction with our attractor model (p. 2, lines 26–30).

References

- Ahmadlou, M, Zweifel, LS, and Heimel, JA. (2018). Functional modulation of primary visual cortex by the superior colliculus in the mouse. *Nat. Commun.* 9.
- Basso, MA, and Wurtz, RH. (1998). Modulation of neuronal activity in superior colliculus by changes in target probability. *J. Neurosci.* 18, 7519–7534.
- Behan, M, Steinhacker, K, Jeffrey-Borger, S, and Meredith, MA. (2002). Chemoarchitecture of GABAergic neurons in the ferret superior colliculus. *J. Comp. Neurol.* 452, 334–359.
- Everling, S, Dorris, MC, Klein, RM, and Munoz, DP. (1999). Role of primate superior colliculus in preparation and execution of anti-saccades and pro-saccades. *J. Neurosci.* 19, 2740–2754.
- Felsen, G, and Mainen, ZF. (2008). Neural Substrates of Sensory-Guided Locomotor Decisions in the Rat Superior Colliculus. *Neuron.* 60, 137–148.
- Felsen, G, and Mainen, ZF. (2012). Midbrain contributions to sensorimotor decision making. *J. Neurophysiol.* 108, 135–147.
- Hirokawa, J, Sadakane, O, Sakata, S, Bosch, M, Sakurai, Y, and Yamamori, T. (2011). Multisensory Information Facilitates Reaction Speed by Enlarging Activity Difference between Superior Colliculus Hemispheres in Rats. *PLoS One.* 6, e25283.
- Horwitz, GD, and Newsome, WT. (2001). Target selection for saccadic eye movements: Direction-selective visual responses in the superior colliculus. *J. Neurophysiol.* 86, 2527–2542.
- Kopec, CD, Erlich, JC, Brunton, BW, Deisseroth, K, and Brody, CD. (2015). Cortical and Subcortical Contributions to Short-Term Memory for Orienting Movements. *Neuron.* 88, 367–377.

- Kopecz, K. (1995). Saccadic reaction times in gap/overlap paradigms: a model based on integration of intentional and visual information on neural, dynamic fields. *Vision Res.* 35, 2911–2925.
- Lafferty, CK, and Britt, JP. (2020). Off-Target Influences of Arch-Mediated Axon Terminal Inhibition on Network Activity and Behavior. *Front. Neural Circuits.* 14, 10.
- Li, N et al. (2019). Spatiotemporal constraints on optogenetic inactivation in cortical circuits. *Elife.* 8.
- Lo, C-C, and Wang, X-J. (2006). Cortico–basal ganglia circuit mechanism for a decision threshold in reaction time tasks. *Nat. Neurosci.* 9, 956–963.
- Mahn, M, Prigge, M, Ron, S, Levy, R, and Yizhar, O. (2016). Biophysical constraints of optogenetic inhibition at presynaptic terminals. *Nat. Neurosci.* 19, 554–556.
- Makin, TR, and De Xivry, JJO. (2019). Ten common statistical mistakes to watch out for when writing or reviewing a manuscript. *Elife.* 8.
- May, PJ. (2006). The mammalian superior colliculus: laminar structure and connections. *Prog. Brain Res.* 151, 321–378.
- Mize, RR. (1992). The organization of GABAergic neurons in the mammalian superior colliculus. *Prog. Brain Res.* 90, 219–248.
- Munoz, DP, and Wurtz, RH. (1995). Saccade-related activity in monkey superior colliculus. I. Characteristics of burst and buildup cells. *J. Neurophysiol.* 73, 2313–2333.
- Najafi, F, Elsayed, GF, Cao, R, Pnevmatikakis, E, Latham, PE, Cunningham, JP, and Churchland, AK. (2020). Excitatory and Inhibitory Subnetworks Are Equally Selective during Decision-Making and Emerge Simultaneously during Learning. *Neuron.* 105, 165-179.e8.
- Salzman, CD, Murasugi, CM, Britten, KH, and Newsome, WT. (1992). Microstimulation in visual area MT: effects on direction discrimination performance. *J. Neurosci.* 12, 2331–2355.
- Sooksawate, T, Isa, K, Behan, M, Yanagawa, Y, and Isa, T. (2011). Organization of GABAergic inhibition in the motor output layer of the superior colliculus. *Eur. J. Neurosci.* 33, 421–432.
- Stubblefield, EA, Costabile, JD, and Felsen, G. (2013). Optogenetic investigation of the role of the superior colliculus in orienting movements. *Behav. Brain Res.* 255, 55–63.
- Thompson, JA, Costabile, JD, and Felsen, G. (2016). Mesencephalic representations of recent experience influence decision making. *Elife.* 5, 1642–1646.
- Trappenberg, TP, Dorris, MC, Munoz, DP, and Klein, RM. (2001). A model of saccade initiation based on the competitive integration of exogenous and endogenous signals in the superior colliculus. *J. Cogn. Neurosci.* 13, 256–271.
- Wang, L, McAlonan, K, Goldstein, S, Gerfen, CR, and Krauzlis, RJ. (2020). A causal role for mouse superior colliculus in visual perceptual decision-making. *J. Neurosci.* 40, 3768–3782.

Reviewers' comments:

Reviewer #2 (Remarks to the Author):

The authors have responded to most of my concerns, I have no further comments.

Reviewer #4 (Remarks to the Author):

The revised manuscript is a substantial improvement over the previous one that we reviewed. We commend the authors on the hard work they have quite clearly put in to overhaul the manuscript significantly. They have taken into account and addressed nearly all of our comments . In this revised ms, it is clear that the authors find an interesting but unexpected result regarding an important question. However, after reading the revised manuscript, and looking carefully at the replotted results, we are still not entirely convinced about the validity of that central result (see "Major points" below). If a convincing case can be made that the unexpected result is 'true' and not the accounted for by confounding factors, that would go a long way towards alleviating our concerns about this result.

Major Points:

1) In Figure 2, which presents the central finding of the ms, we appreciate that the authors have plotted the results per mouse, and displayed the results for each session in each mouse, as this is very useful for looking at the variability in the results- both across mice and across sessions within a mouse. In Figure 2B, which is the main (and unexpected) result of the paper, after a closer look, two things stand out to us, 1) high variability in the results across mice (diamonds) and, 2) high variability in the behavioral results for each mouse, across sessions (grey dots). Mice 9-11 show significant contralateral bias upon photostimulation, followed by 5-8 which seem to potentially show some contralateral bias. Four of the mice (1-4) show no effect.

a. Issue of variability across sessions:

(i) We say 'potentially' for mice 5-8 because in two of these mice (6 and 7), it seems quite likely that the significance is being driven by a single outlier session (to the far right; one of which is shown as the example panel Fig 2A). We suggest that the authors redo this analysis after discarding outlier sessions. If the outcome then becomes that 6/11 mice do not show the effect (1-4, 6 and 7) and only 5 do (9-11, 5 and 8), then the statement "majority of the mice" show the effect (line 170) will no longer be valid. What will this mean for the veracity/reliability of the result?

(ii) Related to this, what might be the reason for this high variability across sessions? Given that the fiber is implanted (fixed) and parameters of stimulation are fixed across sessions, the high variability across sessions (within many mice) is puzzling, and at the very least, needs to be discussed in Discussion.

(iii) And in order to understand the variability in the optogenetic manipulation results, it is also important to contrast it against the variability in the 'control' experiment. The authors should replot the results of Supplementary Figure 2, per mouse, similar to the results of Figure 2B. This will allow an assessment of intrinsic behavioral variability versus variability specifically due to opto manipulation.

b. Issue of variability across mice:

(i) In the methods (lines 728-730), the authors state that the virus was injected 3.64-4.04 mm posterior to bregma, 0.75-1.25 mm lateral of midline, 0.85-2.17 mm dorsal from the brain surface. This is substantial variability in the location of injection across the mouse SC, especially along the dorsal-ventral axis. This variability in injections is critical because it is well known that the distribution of GABA expressing neurons, change with layers and depth (see Lee and Hall, 2006; Isa and Behan, 2011; Villalobos et al 2018). We, therefore, recommend that in Fig. 2, the authors plot the significance of the Bmanipulation, for each mouse as a function of the location of the injection in the SCid (whether it is in SCi or SCd). This would be important as it would allow us to understand

if there are any systematic differences in the effect of stimulating GABAergic neurons according to injection location (especially depth).

(ii) Additionally, to the best of our knowledge, 0.85 mm from the surface of the brain is too shallow to reach the intermediate layer of the SC at any of the AP or ML coordinates listed, and it likely that that injection was actually not in SCi or SCd. We urge the authors to recheck this result and correct us if we have misunderstood their writing.

Consequently, it will be important for their central claim about SCid GABAergic neurons to know where the injections (or at least targets) likely were in each mouse. The authors should present this information as a supplementary table or a figure to make things clear for each of the 11 injections.

2) In Figure 2, (each panel), the authors have used a permutation test the effect of optogenetic manipulation within each mouse, however, we did not find a statistical test which does the same for the mean Bmanipulation, across mice.

Minor points:

1) The authors cite the Horowitz and Newsome (2001) study as evidence for cells in the SC with enhanced firing rates to ipsilateral targets and further mention that “these data are not accounted for by current SC circuit-level frameworks of spatial choice – line 68”. However, we would like to bring to the authors attention that especially in the case of the Horowitz and Newsome study, those authors clarify that such cells are only found in primates that are highly overtrained, as opposed to an expectation that such cells are present under ‘typical’ circumstances. In their follow-up study, Horowitz et al, 2004 specifically answer this question: they find that in a task where direction of motion of the stimulus is dissociated from the saccade direction, they do not find such paradoxical ipsilateral tuning of cells, thus providing evidence that the previous results (Horowitz and Newsome, 2001) were mainly a result of overtraining of animals in a task where the target for ‘action selection’ is not dissociated from the target of ‘spatial choice’.

We suggest that the authors remove line 68 and/or this particular reference or give an explanation regarding this in the discussion.

2) We appreciate the discussion regarding the Wang et al, 2020 study as it brings out important differences between the two studies, however we disagree with the authors that the Wang et al study did not require spatial choice. The task in the Wang et al study is a spatial attention task where the mice had to ignore the distracting stimuli at one spatial location while attending to the cued target at a competing location to report a change in orientation. By definition, that is a spatial selection task. What is possible is: that represented spatial target selection, whereas the current task (in this paper) is an action selection task (as opposed to spatial target selection). We do not agree with the authors that having a 2AFC task in the Wang study (with spatial choice (target selection) and behavioral report (action selection) still dissociated), would have changed those results. However, their other points, about how effects on perception through a couple of different anatomical routes might account for the Wang results being contrary to the current results, are plausible.

3) In this context, we suggest that the title be further modified to “Inhibitory neurons in the superior colliculus mediate action selection” (rather than the very broad ‘goal-directed decision making’).

The points we mention in #2 (Minor) above can be added to the Discussion, as well as introduced in the Intro, thereby clarifying the ‘action selection’. (McPeck and Keller 2004 Nat Neurosci, can be a good source of the appropriate language for this, as they also deal with action selection or saccadic target selection).

We again thank the Reviewers for considering our revised manuscript and for their constructive comments. We have addressed all concerns and have indicated the locations of significant changes in the revised manuscript with page and line numbers, and with crossed-out and red underlined text in the “red-lined” version of the manuscript.

Reviewer #4

The revised manuscript is a substantial improvement over the previous one that we reviewed. We commend the authors on the hard work they have quite clearly put in to overhaul the manuscript significantly. They have taken into account and addressed nearly all of our comments. In this revised ms, it is clear that the authors find an interesting but unexpected result regarding an important question. However, after reading the revised manuscript, and looking carefully at the replotted results, we are still not entirely convinced about the validity of that central result (see “Major points” below). If a convincing case can be made that the unexpected result is ‘true’ and not the accounted for by confounding factors, that would go a long way towards alleviating our concerns about this result.

Major Points:

1. In Figure 2, which presents the central finding of the ms, we appreciate that the authors have plotted the results per mouse, and displayed the results for each session in each mouse, as this is very useful for looking at the variability in the results- both across mice and across sessions within a mouse. In Figure 2B, which is the main (and unexpected) result of the paper, after a closer look, two things stand out to us, 1) high variability in the results across mice (diamonds) and, 2) high variability in the behavioral results for each mouse, across sessions (grey dots). Mice 9-11 show significant contralateral bias upon photostimulation, followed by 5-8 which seem to potentially show some contralateral bias. Four of the mice (1-4) show no effect.

Issue of variability across sessions:

We say ‘potentially’ for mice 5-8 because in two of these mice (6 and 7), it seems quite likely that the significance is being driven by a single outlier session (to the far right; one of which is shown as the example panel Fig 2A). We suggest that the authors redo this analysis after discarding outlier sessions. If the outcome then becomes that 6/11 mice do not show the effect (1-4, 6 and 7) and only 5 do (9-11, 5 and 8), then the statement “majority of the mice” show the effect (line 170) will no longer be valid. What will this mean for the veracity/reliability of the result?

We have reanalyzed our photoactivation data sets with any potential outliers removed. We first determined which sessions could be outliers using the default parameters of the boxplot function in MATLAB (Reviewer Fig. 1). This analysis identified the sessions noted in mice 6 and 7 (Fig. 2b) as well as several other sessions. After excluding these sessions from our analysis, we found that 8/11 Gad2-ChR2 mice showed a significant effect of photoactivation on choice bias ($p < 0.05$). Importantly, the data for mice 6 and 7 remained significant (mouse 4 was the additional mouse that exhibited a significant choice bias with potential outliers removed).

However, we believe it is more conservative to not eliminate sessions that may potentially be outliers, particularly since the variability across sessions is reasonable (see response to Comment 2). In addition, the values shown by mouse (Fig. 2b, diamonds) are not means across sessions, rather they are the values of $\beta_{manipulation}$ when all trials from that mouse are pooled together (similar to the analysis in (Wang *et al.*, 2020)). Thus, even though the effect is even stronger with potential outlier sessions removed, we have retained all sessions in the revised manuscript.

Reviewer Figure 1: Boxplot of photoactivation sessions for each ChR2-expressing Gad2-Cre mouse (as in Fig. 2b). Outliers were identified using the default parameters of the boxplot function in MATLAB and are defined as values that are 1.5 times the interquartile range away from the bottom or top of the box. +, outlier.

2. Related to this, what might be the reason for this high variability across sessions? Given that the fiber is implanted (fixed) and parameters of stimulation are fixed across sessions, the high variability across sessions (within many mice) is puzzling, and at the very least, needs to be discussed in Discussion.

We have clarified in the revised manuscript that the fiber was not fixed in place (p. 30, lines 720–722) and that the parameters of stimulation were not fixed across sessions in our experiments in Gad2-ChR2 mice (Supp. Table 3), both of which may have contributed to variability within mice across sessions (Fig. 2b).

First, the fiber was not fixed in place. Instead, we intentionally stimulated different populations of GABAergic neurons in different sessions by advancing the optetrode through the SC between sessions (p. 30, lines 720–722). Thus, the distance between the fiber tip and the center of ChR2-expressing neurons varied across sessions. We now examine the dependence of choice bias on estimated depth (see response to Comment 4).

Second, the parameters of light stimulation were not fixed across sessions (Supp. Table 3). Instead, within some mice, we used a range of levels of intended light power. However, we did not find an overall dependence of choice bias on intended light power in our Gad2-ChR2 mice (Supp. Fig. 10a). Instead, it is possible that coupling efficiency, and therefore the effective power of light delivered to the brain, varied across sessions, since prior to each session the fiber implanted in the brain was recoupled to the fiber carrying light from the laser (p. 29, lines 707–708). It is possible that different effective light powers may have contributed to the variability across sessions.

3. And in order to understand the variability in the optogenetic manipulation results, it is also important to contrast it against the variability in the ‘control’ experiment. The authors should replot the results of Supplementary Figure 2, per mouse, similar to the results of Figure 2B. This will allow an assessment of intrinsic behavioral variability versus variability specifically due to opto manipulation.

As discussed above (see response to Comment 2) and in our previous response to the reviewers’ comments, due to known sources of variability in experimental conditions across sessions (e.g., different populations of

neurons stimulated), there is a strong rationale for treating each session, rather than mouse, as the unit of analysis (Makin and De Xivry, 2019). For our Gad2-YFP control experiments, we therefore prioritized obtaining data from a sufficient number of sessions (63) rather than mice (2) (Supp. Table 3). Thus, we believe it is most informative to show the comparison between test and control data by session, as we do in Supp. Fig. 2a, rather than mice.

Nevertheless, we performed the same mouse-by-mouse analysis on these 2 control mice as we did on our Gad2-ChR2 test mice. We found that 1 control mouse exhibited a small choice bias ($\beta_{\text{manipulation}} = 0.14$, $p < 0.01$) and 1 did not ($\beta_{\text{manipulation}} = 0.067$, $p = 0.11$). The small choice bias may reflect experimental variability or a very weak off-target effect in this mouse (e.g., activating neurons via heat; p. 30, lines 722–726). Importantly, the effect observed in the control mouse was smaller than that of any of the Gad2-ChR2 mice that exhibited a choice bias (lowest significant $\beta_{\text{manipulation}}$ magnitude = 0.227 (Fig. 2b, mouse 5)), and therefore does not explain our experimental results. We clarify in the revised manuscript that the effect in test mice differed from the effect exhibited by control mice expressing only YFP (p. 8, lines 172–174). These results were unchanged when we reanalyzed the data with potential outliers removed (see response to Comment 1).

Issue of variability across mice:

4. In the methods (lines 728-730), the authors state that the virus was injected 3.64-4.04 mm posterior to bregma, 0.75-1.25 mm lateral of midline, 0.85-2.17 mm dorsal from the brain surface. This is substantial variability in the location of injection across the mouse SC, especially along the dorsal-ventral axis. This variability in injections is critical because it is well known that the distribution of GABA expressing neurons, change with layers and depth (see Lee and Hall, 2006; Isa and Behan, 2011; Villalobos et al 2018). We, therefore, recommend that in Fig. 2, the authors plot the significance of the $\beta_{\text{manipulation}}$, for each mouse as a function of the location of the injection in the SCid (whether it is in SCi or SCd). This would be important as it would allow us to understand if there are any systematic differences in the effect of stimulating GABAergic neurons according to injection location (especially depth).

The reviewer is correct that there was substantial variability in the depth of our injections. In each mouse, we made two or three injections to cover the top and middle of the SCid; the depths chosen for each mouse depended on the rostral-caudal coordinates of the injection site. Importantly, we only analyzed data obtained from the intermediate and deep layers which we determined based on the placement of the fiber tip. Thus, the stronger determinant of stimulation depth was not the injection site, but the depth of the fiber, which we varied to photoactivate different populations of intermediate and deep layer GABAergic SC neurons by advancing the optrode ventrally between sessions (see response to Comment 2). In the revised manuscript, we have examined the effect of depth on choice bias in New Supplementary Fig. 9b and found that the magnitude of $\beta_{\text{manipulation}}$ increased with increased depth ($p = 0.0011$, Pearson correlation).

5. Additionally, to the best of our knowledge, 0.85 mm from the surface of the brain is too shallow to reach the intermediate layer of the SC at any of the AP or ML coordinates listed, and it likely that that injection was actually not in SCi or SCd. We urge the authors to recheck this result and correct us if we have misunderstood their writing. Consequently, it will be important for their central claim about SCid GABAergic neurons to know where the injections (or at least targets) likely were in each mouse. The authors should present this information as a supplementary table or a figure to make things clear for each of the 11 injections.

While there was some variability in the relative strength of ChR2-YFP expression between the intermediate and deep layers across mice, we verified opsin expression in these layers in every mouse (see examples in Supp. Fig. 1). Critically, although there may have been some ChR2-YFP expression in the superficial layers in some mice, we ensured that we did not photoactivate superficial-layer SC neurons by targeting the depth of the fiber tip to the intermediate and deep layers. We estimated these depths, across sessions, as 0.32-1.2 mm from the surface of the SC (see response to Comments 2 and 4).

6. In Figure 2, (each panel), the authors have used a permutation test the effect of optogenetic manipulation

within each mouse, however, we did not find a statistical test which does the same for the mean $B_{manipulation}$, across mice.

In the revised manuscript, we report that the $\beta_{manipulation}$ values across mice (Fig. 2b, diamonds), were significantly different from 0 ($p = 0.0029$, Wilcoxon signed-rank test; p. 7-8, lines 170–171).

Minor points:

7. The authors cite the Horowitz and Newsome (2001) study as evidence for cells in the SC with enhanced firing rates to ipsilateral targets and further mention that “these data are not accounted for by current SC circuit-level frameworks of spatial choice – line 68)”. However, we would like to bring to the authors attention that especially in the case of the Horowitz and Newsome study, those authors clarify that such cells are only found in primates that are highly overly trained, as opposed to an expectation that such cells are present under ‘typical’ circumstances. In their follow-up study, Horowitz et al, 2004 specifically answer this question: they find that in a task where direction of motion of the stimulus is dissociated from the saccade direction, they do not find such paradoxical ipsilateral tuning of cells, thus providing evidence that the previous results (Horowitz and Newsome, 2001) were mainly a result of overtraining of animals in a task where the target for ‘action selection’ is not dissociated from the target of ‘spatial choice’. We suggest that the authors remove line 68 and/or this particular reference or give an explanation regarding this in the discussion.

We have removed the reference to Horowitz and Newsome (2001) in this section of the revised Introduction and in the Discussion.

8. We appreciate the discussion regarding the Wang et al, 2020 study as it brings out important differences between the two studies, however we disagree with the authors that the Wang et al study did not require spatial choice. The task in the Wang et al study is a spatial attention task where the mice had to ignore the distracting stimuli at one spatial location while attending to the cued target at a competing location to report a change in orientation. By definition, that is a spatial selection task. What is possible is: that represented spatial target selection, whereas the current task (in this paper) is an action selection task (as opposed to spatial target selection). We do not agree with the authors that having a 2AFC task in the Wang study (with spatial choice (target selection) and behavioral report (action selection) still dissociated), would have changed those results. However, their other points, about how effects on perception through a couple of different anatomical routes might account for the Wang results being contrary to the current results, are plausible.

We have clarified in the revised Discussion that the task used by Wang et al. (2020) did not require the selection of a spatial target for a movement (whereas ours did), which may have contributed to the difference between our results and that of Wang et al. (p. 23, lines 566–568). We have further clarified that additional experiments would be required to determine whether this difference in task requirements was responsible for the different results across studies (p. 23-24, lines 568–571).

9. In this context, we suggest that the title be further modified to “Inhibitory neurons in the superior colliculus mediate action selection” (rather than the very broad ‘goal-directed decision making’). The points we mention in #2 (Minor) above can be added to the Discussion, as well as introduced in the Intro, thereby clarifying the ‘action selection’. (McPeck and Keller 2004 Nat Neurosci, can be a good source of the appropriate language for this, as they also deal with action selection or saccadic target selection).

We appreciate the reviewer’s suggestion as well as the distinction between action selection and spatial choice. However, we are concerned that the phrase “action selection” would make our paper less accessible to a broad readership for whom the term may be unfamiliar, and that it carries implications that we do not address here. However, we have added “spatial” to the revised title to specify the form of decision making that we directly study. In addition, we have made the link between action selection and spatial choice explicit by revising our definition of spatial choice in the Results to “the process of selecting either the left or right reward location for movement (i.e., action selection)” (p. 5-6, lines 122–124).

References

Makin, TR, and De Xivry, JJO. (2019). Ten common statistical mistakes to watch out for when writing or reviewing a manuscript. *Elife*. 8.

Wang, L, McAlonan, K, Goldstein, S, Gerfen, CR, and Krauzlis, RJ. (2020). A causal role for mouse superior colliculus in visual perceptual decision-making. *J. Neurosci.* 40, 3768–3782.

REVIEWERS' COMMENTS:

Reviewer #4 (Remarks to the Author):

The authors have done a good job addressing (nearly) all of our concerns. There is just one remaining concern, and this is one that we brought up last time (point 9 in the authors' rebuttal).

Referring to the behavior in this paper as 'spatial decision-making' in the title is potentially misleading. The definition of spatial choice in the revised Results as "the process of selecting either the left or right reward location for movement (i.e., action selection)" (revised lines 122-124) does not adequately address this issue – it was not sufficiently clarifying on its own, and the title itself would still be potentially misleading.

Historically, when locations of the stimuli between which a choice is to be made are not separate from the locations of the behavioral reports, referring to such behavior as spatial 'decision-making' is problematic. The monkey SC literature has grappled with this issue, with such selection referred to as 'saccade target selection' or 'saccade selection' (or in other words, 'action selection'). This is an important conceptual issue; a dissociation of sensory locations from motor plan/behavioral locations would be necessary, in our opinion, for the unambiguous use of 'spatial decision-making'. Moreover, the term 'action selection' is also used visibly in the literature - two very recent examples just from mouse SC work are Huda et al, 2019; Eur J Neurosci; Huda et al, 2020, Nat Commun. We disagree that 'action selection' would make the paper any less accessible, and argue that accuracy should drive the choice of terminology. We, therefore, urge the authors strongly to consider using something akin to 'saccade target selection' in the title, rather than 'spatial decision-making', and then update revised lines 122-124 appropriately.